# Myelination synchronizes cortical oscillations by consolidating parvalbumin-mediated phasic inhibition

**Mohit Dubey[1], Maria Pascual-Garcia[2], Koke Helmes[1], Dennis D Wever[1], Mustafa S Hamada[1,3], Steven A Kushner[2], Maarten HP Kole[1,3]\***

[1]Department of Axonal Signaling, Netherlands Institute for Neuroscience (NIN), Royal Netherlands Academy of Arts and Sciences (KNAW), Amsterdam, Netherlands; [2]Department of Psychiatry, Erasmus Medical Centre, Rotterdam, Netherlands; [3]Cell Biology, Neurobiology and Biophysics, Department of Biology, Faculty of Science, Utrecht University, Utrecht, Netherlands

**Abstract** Parvalbumin-positive (PV[+]) γ-aminobutyric acid (GABA) interneurons are critically involved in producing rapid network oscillations and cortical microcircuit computations, but the significance of PV[+] axon myelination to the temporal features of inhibition remains elusive. Here, using toxic and genetic mouse models of demyelination and dysmyelination, respectively, we find that loss of compact myelin reduces PV[+] interneuron presynaptic terminals and increases failures, and the weak phasic inhibition of pyramidal neurons abolishes optogenetically driven gamma oscillations in vivo. Strikingly, during behaviors of quiet wakefulness selectively theta rhythms are amplified and accompanied by highly synchronized interictal epileptic discharges. In support of a causal role of impaired PV-mediated inhibition, optogenetic activation of myelin-deficient PV[+] interneurons attenuated the power of slow theta rhythms and limited interictal spike occurrence. Thus, myelination of PV axons is required to consolidate fast inhibition of pyramidal neurons and enable behavioral state-dependent modulation of local circuit synchronization.

**\*For correspondence:** m.kole@nin.knaw.nl

**Competing interest:** The authors declare that no competing interests exist.

## Editor's evaluation

This study shows that demyelination results in reduced inhibition by decreasing the number of release sites for the inhibitory transmitter GABA, thus promoting epileptic activity. Importantly, the paper describes that demyelination causes an increase in theta oscillation power during quiet wakefulness and an impairment of optogenetically induced γ oscillations in the cortex. These results expand our understanding of the functions of myelin in gray matter and its clinical relevance to demyelinating disorders such as multiple sclerosis.

## Introduction

GABAergic interneurons play fundamental roles in controlling rhythmic activity patterns and the computational features of cortical circuits. Nearly half of the interneuron population in the neocortex is parvalbumin-positive (PV[+]) and comprised mostly of the basket cell (BC) type (*Hu et al., 2014*; *Tremblay et al., 2016*). PV[+] BCs are strongly and reciprocally connected with pyramidal neurons (PNs) and other interneurons, producing temporally precise and fast inhibition (*Bartos et al., 2002*; *Gonchar and Burkhalter, 1997*; *Tamás et al., 1997*). The computational operations of PV[+] BCs, increasing gain control, sharpness of orientation selectively, and feature selection in the sensory cortex (*Atallah et al., 2012*; *Cardin et al., 2009*; *Lee et al., 2012*; *Yang et al., 2017*; *Zucca et al., 2017*), are mediated by

**eLife digest** The brain contains billions of neurons that connect with each other via cable-like structures called axons. Axons transmit electrical impulses and are often wrapped in a fatty substance called myelin. This insulation increases the speed of nerve impulses and reduces the energy lost over long distances. Loss or damage of the myelin layer – as is the case for multiple sclerosis, a chronic neuroinflammatory and neurodegenerative disease of the central nervous system – can cause serious disability.

However, a fast-firing neuron within the brain, called PV$^+$ interneuron, has short, sparsely myelinated axons. Even so, PV$^+$ interneurons are powerful inhibitors that regulate important cognitive processes in gray matter areas, including the outermost parts, in the cortex. Yet it remains unclear how the unusual, patchy myelination affects their function.

To examine these questions, Dubey et al. used genetically engineered mice either lacking or losing myelin and studied the impact on PV$^+$ interneurons and slow brain waves. As mice progressively lost myelin, the speed of inhibitory signals from PV$^+$ interneurons did not change but their signal strength decreased. As a result, the power of slow brain waves, no longer inhibited by PV$^+$ interneurons, increased. These waves also triggered spikes of epileptic-like brain activity when the mice were inactive and quiet. Restoring the activity of myelin-deficient PV$^+$ interneurons helped to reverse these deficits.

This suggests that myelination, however patchy on PV$^+$ interneurons, is required to reach their full inhibitory potential. Moreover, the findings shed light on how myelin loss might underpin aberrant brain activity, which have been observed in people with multiple sclerosis. More research could help determine whether these epilepsy-like spikes could be a biomarker of multiple sclerosis and/or a target for developing new therapeutic strategies to limit cognitive impairments.

a range of unique molecular and cellular specializations. Their extensive axon collaterals targeting hundreds of PNs are anatomically arranged around the soma and dendrites, and electrotonically close to the axonal output site. In addition, the unique calcium (Ca$^{2+}$) sensor in PV$^+$ BCs terminals, synaptotagmin 2 (Syt2), is tightly coupled to Ca$^{2+}$ channels mediating fast and synchronized release kinetics (*Chen et al., 2017*; *Sommeijer and Levelt, 2012*), powerfully shunting excitatory inputs and increasing the temporal precision of spike output (*Hu et al., 2014*; *Somogyi et al., 1983*; *Tamás et al., 1997*; *Thomson et al., 1996*).

Recent findings have shown that the proximal axons of PV$^+$ interneurons are covered by myelin sheaths (*Micheva et al., 2016*; *Peters and Proskauer, 1980*; *Somogyi et al., 1983*; *Stedehouder et al., 2017*; *Tamás et al., 1997*; *Yang et al., 2020*). How interneuron myelination defines cortical inhibition remains, however, still poorly understood. Myelination of axons provides critical support for long-range signaling by reducing the local capacitance that results in rapid saltatory conduction and by maintaining the axonal metabolic integrity (*Cohen et al., 2020*; *Nave and Werner, 2014*). For PV$^+$ BCs, however, the average path length between the axon initial segment (AIS) and release sites involved in local circuit inhibition is typically less than ~200 μm (*Micheva et al., 2021*; *Schmidt et al., 2017*; *Tamás et al., 1997*), and theoretical and experimental studies indicate the acceleration by myelin may play only a limited role (*Micheva et al., 2021*; *Micheva et al., 2016*). Another notable long-standing hypothesis is that myelination of PV$^+$ axons may be critical for the security and synchronous invasion of presynaptic terminals (*Somogyi et al., 1983*). In support of a role in reliability, in Purkinje cell axons of the long Evans shaker (*les*) rat, which carries a deletion of *Mbp*, spike propagation shows failures and presynaptic terminals are disrupted (*Barron et al., 2018*). Interestingly, in a genetic model in which oligodendrocyte precursor cells lack the γ2 GABA$_A$ receptor subunit, fast-spiking interneuron axons in the neocortex are aberrantly myelinated and feedforward inhibition is impaired (*Benamer et al., 2020*). At the network level, PV$^+$ BC-mediated feedback and feedforward inhibition is critical to produce local synchronization between PNs and interneuron at the gamma (γ) frequency (30–80 Hz), which is a key rhythm binding information from cell assemblies, allowing synaptic plasticity and higher cognitive processing of sensory information (*Buzsáki, 2006*; *Cardin et al., 2009*; *Hu et al., 2014*; *Sohal et al., 2009*; *Veit et al., 2017*). Here, we determined whether PV$^+$ BC-driven neocortical rhythms require myelination by using de- and dysmyelination models, studying

the cellular properties of genetically labeled PV[+] BCs and examining the functional role of myelin by longitudinally examining the frequency spectrum of cortical oscillations.

## Results

### Behavioral state-dependent increase in theta power and interictal epileptiform discharges

We investigated in vivo cortical rhythms by recording local field potential (LFP) in layer 5 (L5) together with surface electrocorticogram (ECoG) signals from both primary somatosensory (S1) and visual (V1) areas. Freely moving mice (C57BL/6, 7–9 weeks at the start of the experiment) were recorded in their home cage every second week (18–24 hr/week) across an 8-week cuprizone treatment, inducing toxic loss of oligodendrocytes in white- and gray matter areas (*Clarner et al., 2012*; *Hamada and Kole, 2015*; *Kipp et al., 2009*). Remarkably, after 6 weeks of cuprizone feeding we detected high-voltage spike discharges (approximately five times the baseline voltage and ~50–300 ms in duration, *Figure 1a–c*). These brief spike episodes on the ECoG and LFP (*Figure 1c*) occurred bilaterally and near synchronously in S1 and V1, resembling the interictal epileptiform discharges (also termed interictal spikes) that are a hallmark of epilepsy (*Cohen et al., 2002*; *Dubey et al., 2018*; *Hoffmann et al., 2008*; *Tóth et al., 2018*). Automated detection of interictal spikes in the raw ECoG–LFP signal was performed with a machine-based learning classifier (see *Figure 1—figure supplement 1* and Materials and methods), revealing a progressively increasing number of interictal spikes, from ~5/hr at 4 weeks up to ~70/hr at 8 weeks of cuprizone treatment (*Figure 1d*). Interestingly, interictal spikes were highly dependent on vigilance state and present exclusively during quiet wakefulness (30 out of 30 randomly selected LFP segments from awake or quiet wakefulness, chi-square test p<0.0001, n = 6 cuprizone mice), with no other discernible association to specific behaviors (*Figure 1—figure supplement 1b*, *Figure 1—video 1* ). Whether the pathological cortical oscillations were specific to certain frequency bands, including gamma (γ, 30–80 Hz), was examined by plotting the power spectrum density of the LFP in S1 during periods of quiet wakefulness or active movement (*Figure 1e*). During quiet wakefulness, LFP power in cuprizone-treated mice was selectively amplified in the theta frequency band (θ, 4–12 Hz, Šidák's multiple comparisons test, p=0.0013, *Figure 1e and f*, *Figure 1—figure supplement 1c*). In contrast, during active states when mice were moving and exploring no differences were observed in the power spectrum, in none of the frequency bands (Šidák's multiple comparisons test, p>0.166, *Figure 1e and f*, *Source data 1*). Finally, to more firmly establish whether interictal epileptiform discharges result from the lack of myelin, we analyzed ECoG signals in the dysmyelinated shiverer mice (*Mbp*[Shi]) lacking compact myelin due to a truncating mutation in *Mbp* (*Readhead et al., 1987*). Shiverer mice suffer progressively increasing number of epileptic seizures beginning at approximately 8 weeks of age (*Chernoff, 1981*; *Readhead et al., 1987*). ECoG recordings at 8 weeks showed that in addition to ictal discharges interictal spikes were detected with a rate of ~1/min, comparable to cuprizone-treated mice (*Figure 1g and h*, *Figure 1—figure supplement 2*). Although the waveform of interictal spikes in shiverer was substantially longer in duration (~100–500 ms), analysis of the power across the four frequency bands around interictal spikes revealed no difference in comparison to the cuprizone-treated mice (two-way analysis of variance [ANOVA] p=0.7875, n = 6 mice for both groups, *Figure 1—figure supplement 2*).

### Loss of myelin impairs fast PV[+] BC-mediated inhibition

Increased power of sensory-driven slow oscillations and epileptiform activity in the neocortex of normally myelinated brains is also observed when PV[+] interneurons are optogenetically silenced (*Brill et al., 2016*; *Veit et al., 2017*; *Yang et al., 2017*). To investigate how myelin loss affects the PV[+] interneuron morphological and functional properties, we crossed the *Pvalb*[Cre] mouse line, having Cre recombinase targeted to *Pvalb*-expressing cells, with a tdTomato fluorescence (Ai14) Cre reporter line (hereafter called PV-Cre; Ai14 mice). The cytoplasmic fluorescence allowed quantification of PV[+] cell bodies and their processes in the primary somatosensory cortex (*Figure 2a and b*, *Figure 2—figure supplement 1a*), and immunofluorescent labeling with myelin basic protein (MBP) revealed substantial myelination of large-diameter PV[+] axons (mean ± SEM, 80.15% ± 9.95% along 83 mm of PV[+] axons analyzed, n = 3 slices from two mice, z-stack with a volume of 7.66 × 10[5] μm[3], *Figure 2b*, *Figure 2—figure supplement 1a*). Electron microscopy (EM) immunogold-labeled tdTomato showed that PV[+]

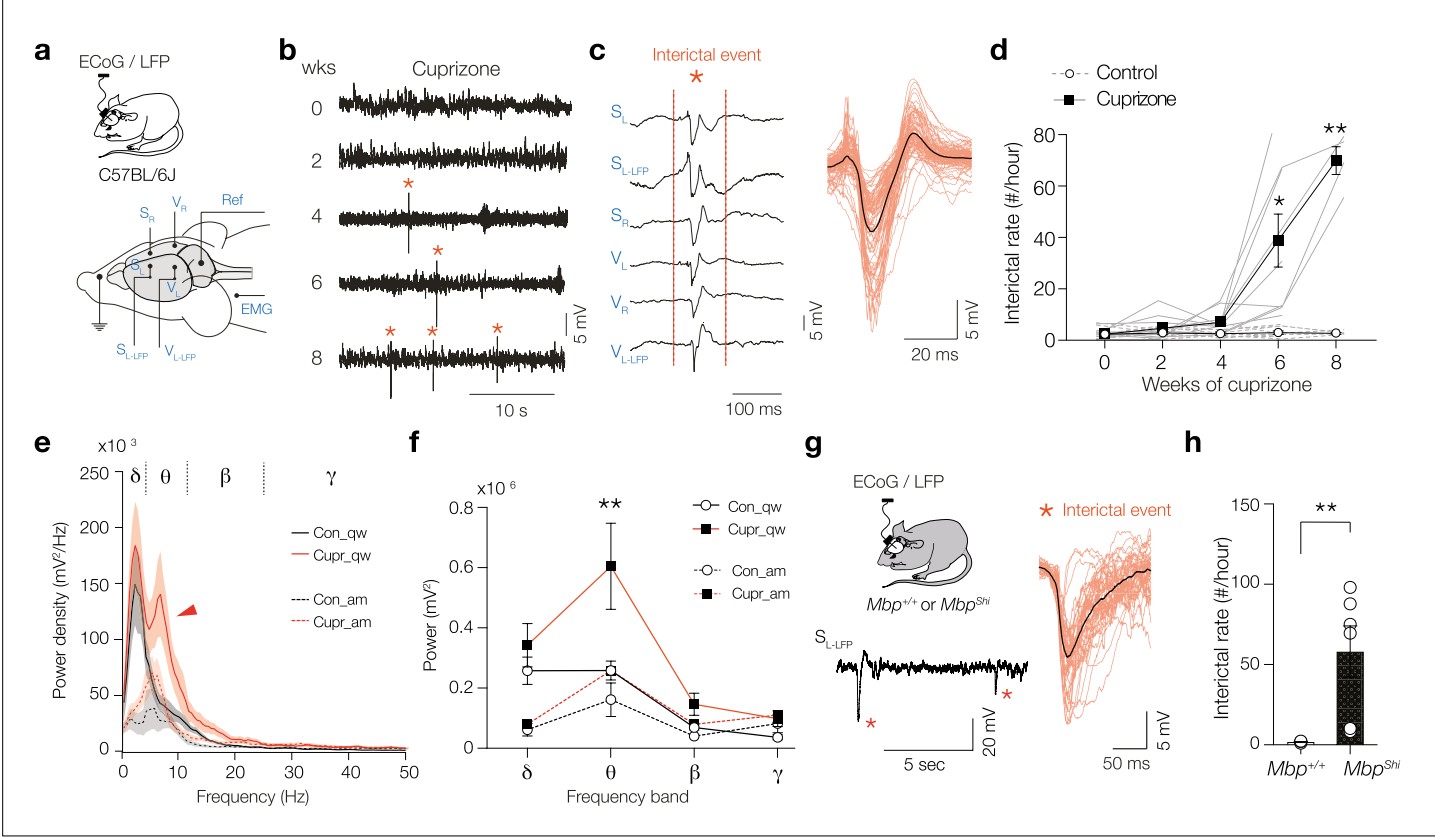

**Figure 1.** Loss of compact myelin causes interictal spikes and behavioral state-dependent amplification of theta rhythms. (**a**) Schematic of the electrocorticogram (ECoG) and local field potential (LFP) recordings in freely moving mice. Electrodes were placed right ($S_R$) and left ($S_L$) in the primary somatosensory cortex, and a left LFP electrode ($S_{L-LFP}$) into layer 5 (L5). A similar array of electrodes was positioned in the primary visual cortex ($V_R$, $V_L$, and $V_{L-LFP}$). One electrode was placed around neck muscle recording electromyography (EMG) and one used as reference (Ref). (**b**) Interictal spikes (*) appear from 4-week cuprizone and onward. Example raw LFP traces ($S_{L-LFP}$). (**c**) Representative interictal spike example showing spatiotemporal synchronization of the spike across cortical areas and hemispheres. Higher magnification of interictal spikes (red, ~50–300 ms duration) overlaid with the average (black). (**d**) Cuprizone treatment caused a progressively increasing frequency of interictal spikes (mixed-effects two-way ANOVA). Time × treatment interaction p<0.0001, Šidák's multiple comparisons tests, cupri vs. con; 4 weeks p=0.121, 6 weeks *p=0.0406 and 8 weeks **p=0.0054. (**e**) Power spectral content during two different brain states, awake and moving (am, dotted lines) and quiet wakefulness (qw, solid lines) in control (black) and cuprizone (red). Red arrow marks amplified theta band power ($\theta$) during quite wakefulness in cuprizone mice (Cupr_qw). (**f**) Cuprizone amplifies selectively $\theta$ power during quiet wakefulness (two-way ANOVA treatment p<0.0001, Šidák's multiple comparisons cupr vs. con; for δ, p=0.8118; $\theta$, p=0.0013; β, p=0.8568 and for γ, p=0.9292) but not during moving (two-way ANOVA treatment p=0.0575). (**g**) Left: schematic of ECoG and LFP recordings from $Mbp^{+/+}$ and $Mbp^{Shi}$ mice with example trace showing interictal spikes (*). Right: overlaid individual interictal spikes in $Mbp^{Shi}$ mice (red) combined with the population average (black). (**h**) Bar plot of interictal rate in $Mbp^{Shi}$ mice. Two-tailed Mann–Whitney test **p=0.0095. Data shown as mean ± SEM with gray lines (**d**) or open circles (**h**) individual mice.

The online version of this article includes the following video and figure supplement(s) for figure 1:

**Figure supplement 1.** State-dependent interictal activity in cuprizone mice and automated interictal event detection library.

**Figure supplement 2.** Ictal, interictal activity, and power spectrum in $Mbp^{Shi}$ mice.

**Figure 1—video 1.** Brain state-dependent interictal spikes.

https://elifesciences.org/articles/73827/figures#fig1video1

axons possessed multilamellar compact myelin sheaths (on average, 6.33 ± 0.80 myelin lamella) with 10.8 ± 0.76 nm distance between the major dense lines and a mean *g*-ratio (axon diameter/fiber diameter) of 0.74 ± 0.01 (n = 6 sheaths, *Figure 2c*). PV-Cre; Ai14 mice fed with 0.2% cuprizone for 6 weeks showed strongly reduced MBP in S1 and PV+ axons were largely devoid of myelin (*Figure 2a and b*, *Figure 2—figure supplement 1b*) while the total number of PV+ cell bodies across cortical layers remained constant (control density, 326 ± 14 cells mm−2 vs. cuprizone density, 290 ± 48 cells mm−2, n = 6 sections from N = 6 animals/group, Mann–Whitney test p=0.1649, *Figure 2d*). Further,

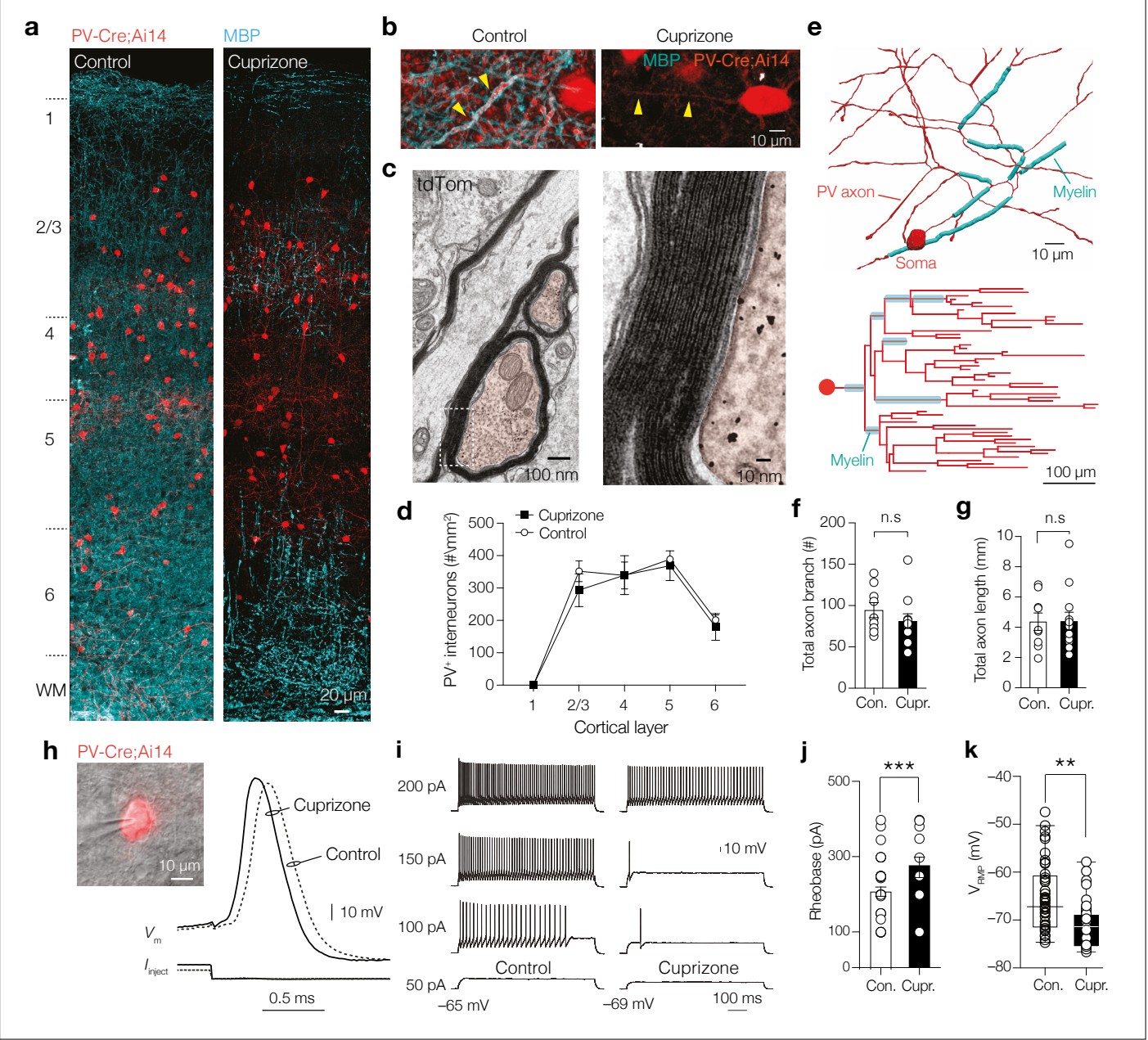

**Figure 2.** Demyelination preserves parvalbumin-positive (PV⁺) interneuron number and morphology but reduces excitability. (**a**) Left: confocal z-projected overview image of S1 in a PV-Cre; Ai14 mouse (tdTomato⁺, red) overlaid with myelin basic protein (MBP, cyan). Right: overview image showing loss of MBP after + weeks cuprizone. (**b**) Myelinated PV⁺ axons in control (left) and PV⁺ axons demyelination with cuprizone treatment (right). (**c**) Electron microscopy (EM) of transverse cut tdTomato⁺ immunogold-labeled axons (false-colored red). Right: higher magnification of immunogold particles and ultrastructure of the PV interneuron myelin sheath. (**d**) PV⁺ interneuron number across the cortical lamina was not affected by cuprizone treatment (two-way ANOVA, treatment effect p=0.6240). (**e**) Top: example of a high-resolution 3D reconstruction of a biocytin-labeled PV axon (red) labeled with MBP (cyan) of a control mouse, showing the first approximately six axonal branch orders. Bottom: control axonogram showing axon branch order and myelinated segments (cyan). (**f, g**) Total axon branch number and length are not changed by demyelination (Mann–Whitney tests p=0.3032 and p=0.8822, respectively). n.s., not significant. (**h**) Left: brightfield/fluorescence overlay showing patch-clamp recording from a tdTomato⁺ interneuron. Right: example PV⁺ interneuron APs from control (dotted line) and cuprizone-treated mice (continuous lines). (**i**) Steady-state sub- and suprathreshold voltage responses during 700 ms current injections. Firing rates near threshold reduced and the (**j**) rheobase current significantly increased in cuprizone (***p=0.0003). (**k**) PV⁺ basket cells (BCs) showed an ~4 mV hyperpolarized resting membrane potential (**p=0.0036). Data shown as mean ± SEM and open circles individual cells.

The online version of this article includes the following figure supplement(s) for figure 2:

**Figure supplement 1.** Cuprizone treatment causes loss of parvalbumin-positive (PV⁺) axon myelination but preserves axon length and complexity.

single-cell analysis was performed on biocytin-filled PV-Cre[+] interneurons that were re-sectioned and stained for MBP to identify the location of myelin and the axon morphology (*Figure 2e*, *Figure 2— figure supplement 1c and d*). Myelin was present on multiple proximal axonal segments of all control BCs (4/4 fully reconstructed axons, on average 2.8% ± 1.2% myelination). In contrast, none of the BCs from cuprizone-treated mice showed myelinated segments (0/6 axons). Furthermore, the total number of axon segments (~80 per axon, Mann–Whitney test p=0.3032, *Figure 2f*) as well as the total path length were unaffected by cuprizone treatment (on average ~4.5 mm in both groups, range 2.0–9.5 mm, Mann–Whitney test p=0.9871, *Figure 2g*, *Figure 2—figure supplement 1c–f*).

To examine whether myelin loss changes the intrinsic excitability of PV[+] BCs, we made whole-cell recordings in slices from PV-Cre; Ai14 mice (*Figure 2h*). Recording of steady-state firing properties by injecting increasing steps of currents injections revealed an increase in the rheobase current (~90 pA, *Figure 2i and j*) and an ~50 Hz reduced firing frequency during low-amplitude current injections (two-way ANOVA, treatment p=0.0441, Šidák's multiple comparisons post hoc test at 200 pA; p=0.0382, 250 pA; p=0.0058, 300 pA; p=0.0085) without a change in the maximum instantaneous firing rate (two-way ANOVA, Šidák's multiple comparisons post hoc test, p=0.92, data not shown). However, neither the AP half-width (control, 290 ± 10 µs, n = 34 cells from 12 mice vs. cuprizone, 295 ± 10 µs, n = 15 cells from 7 mice, two-tailed Mann–Whitney $U$-test p=0.7113) nor AP amplitude was affected by cuprizone treatment (control 78.12 ± 1.66 mV, n = 34 cells from 12 mice vs. cuprizone, 80.13 ± 2.48 mV, n = 15 cells from 7 mice, p=0.4358). In contrast, the resting membrane potential ($V_{RMP}$) of PV[+] interneurons was on average ~4 mV significantly more hyperpolarized (*Figure 2k*) without a change in the apparent input resistance (control, 133.3 ± 8.55 MΩ, n = 42 cells from 21 mice vs. cuprizone 125 ± 8.96 MΩ, 27 cells out of 13 mice, p=0.5952). In addition to the hyperpolarization in $V_{RMP}$, demyelinated PV[+] interneurons also had an ~3 mV more hyperpolarized AP voltage threshold (control, –40.51 ± 0.97 mV, n = 34 cells from 12 mice) and cuprizone –43.65 ± 1.29 (n = 15 cells from 7 mice, Mann–Whitney test p=0.0269). Taken together, the results indicate that cuprizone treatment completely demyelinates proximal branches of PV[+] interneuron axons, and while not affecting axon morphology, causes a net decrease in the intrinsic PV[+] interneuron excitability.

Is myelin required for PV[+] BC-mediated inhibition? Single PV[+] BCs typically make 5–15 synapses with a PN in a range of <200 µm, forming highly reliable, fast, and synchronized release sites (*Micheva et al., 2021*; *Packer and Yuste, 2011*; *Tamás et al., 1997*; *Thomson et al., 1996*). Intercellular variations in both myelin distribution and aberrant myelin patterns have been associated with conduction velocity changes (*Benamer et al., 2020*; *Micheva et al., 2021*). To examine the role of myelin on inhibitory transmission, we made paired recordings of PV[+] BCs and L5 PNs with and without myelination, in control or cuprizone-treated PV-Cre; Ai14 mice, respectively (*Figure 3*). We evoked APs in PV[+] BCs while recording unitary inhibitory postsynaptic currents (uIPSCs) under conditions of physiological $Ca^{2+}$/ $Mg^{2+}$ (2.0/1.0 mM in n = 78 pairs, *Figure 3a and b*). Concordant with optogenetic mapping of PV[+] inputs onto L5 PNs in mouse S1 (*Packer and Yuste, 2011*), in control slices the probability of a given PV[+] cell being connected to a nearby PN was high (~0.48, *Figure 3c*). In contrast, the connection probability was significantly lower in cuprizone-treated mice (~0.23, p=0.0182, *Figure 3b and c*). In 13 stable connected pairs, we examined unitary IPSC properties including failure rate and amplitude, as well as rise- and decay time, using automated fits of the uIPSCs (n > 80 trials per connection, *Figure 3d*). Cuprizone treatment led to a significant increase in the number of failures (from 0.05 to 0.26, *Figure 3e*) and an ~2.5-fold reduction in the average uIPSC peak amplitude (*Figure 3f*). To obtain an estimate of propagation speed, we determined on successful trials the latency between the AP peak and uIPSCs at 10% peak amplitude (*Figure 3d*). Interestingly, both the mean latency and the trial-to-trial latency variability remained unchanged (average ~800 µs; Mann–Whitney test p>0.999; SD in cuprizone 319 ± 65 µs, n = 7 pairs, SD in control, 276 ± 38 µs, n = 5 pairs, p>0.60, *Figure 3g*).

To further examine the properties of GABA release in demyelinated PV-BCs, we recorded uIPSCs during a train of five APs at 100 Hz (averaging >50 trials, *Figure 3h*). Consistent with the temporary facilitation in IPSCs of adult Purkinje cells (*Turecek et al., 2016*), uIPSC recordings in control PV BCs showed that paired-pulse ratios were on the second spike facilitated by 20% (uIPSC$_2$/uIPSC$_1$ 1.20 ± 0.060) and gradually depressed on the subsequent spikes (spikes 3–5). In contrast, in cuprizone-treated mice uIPSCs were depressed during the second and subsequent pulses (two-way repeated-measures ANOVA pulse × treatment effect p<0.021, Šidák's multiple comparisons tests for uIPSC$_2$/uIPSC$_1$ 0.89 ± 0.041, p=0.0339, *Figure 3h*).

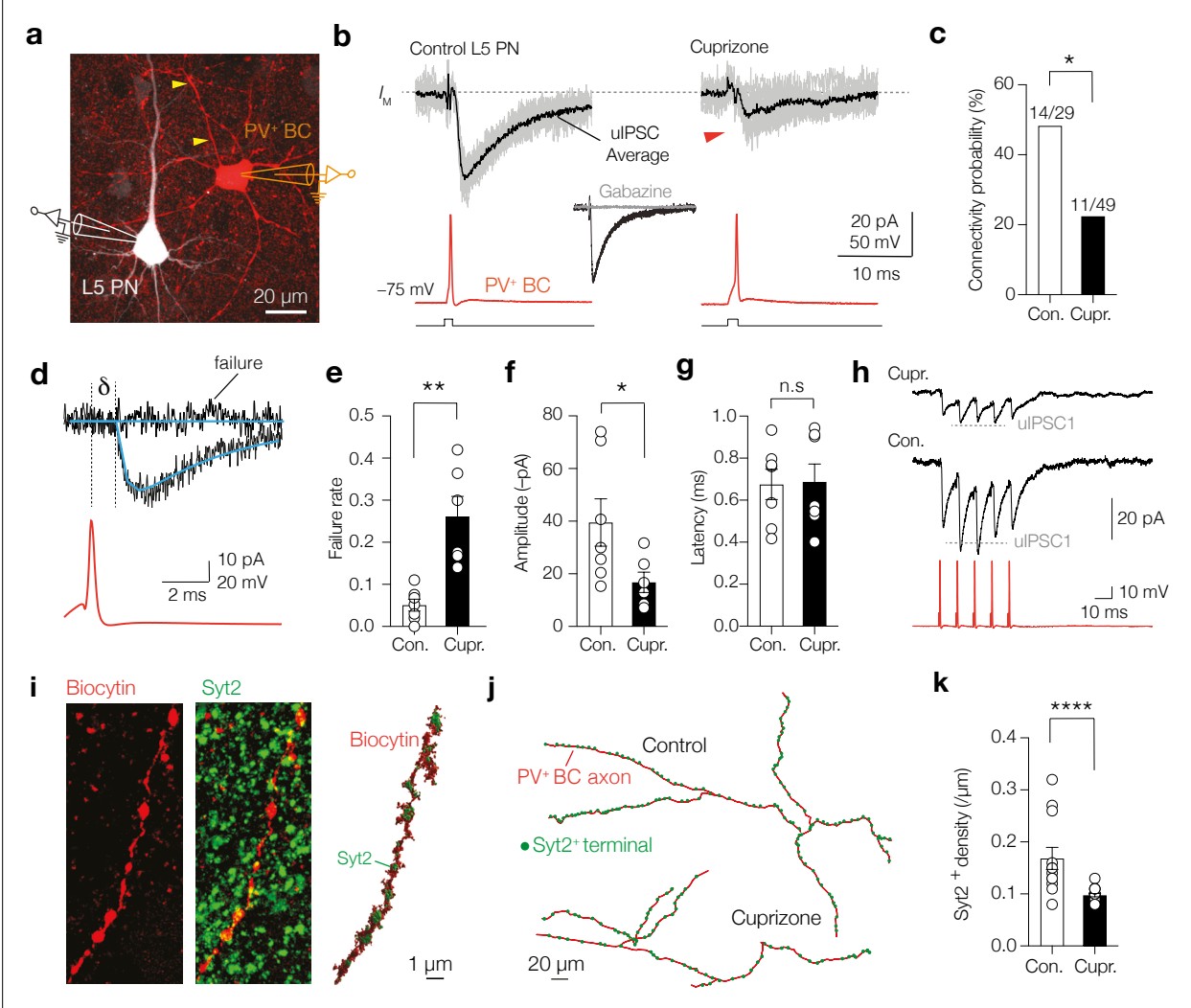

**Figure 3.** Demyelination decreases connectivity, reliability, and rapid facilitation of parvalbumin-positive (PV+) unitary inhibitory postsynaptic currents (uIPSCs). (**a**) Immunofluorescence image of a connected control PV+ basket cell (BC) (red) and layer 5 (L5) pyramidal neuron (PN) (white). (**b**) Example traces of 10 single-trial uIPSC traces (gray) overlaid with mean average (>60 trials, black). Inset: uIPSCs abolished by gabazine (GABA$_A$ blocker, 4 µM). (**c**) Cuprizone-treated mice show significantly lower connection probability between PV+ BC and L5-PN (chi-square test *p=0.0182, n = 78 pairs). (**d**) Example fits (blue) of uIPSCs for rise- and decay time, amplitude, failure rate, amplitude, and latency analyses (δ, AP to 10% uIPSC peak amplitude). (**e**) Cuprizone increased failures by fivefold and the average peak amplitude by ~2.5-fold (Mann–Whitney test **p=0.012) and (**f**) reduces the mean amplitude (Mann–Whitney test *p=0.0256). (**g**) uIPSCs latency remained unchanged (Mann–Whitney test n.s., p>0.999). (**h**) Cuprizone impairs short-term facilitation. Dotted line indicates expected amplitude for uIPSC$_2$ (scaled from uIPSC$_1$). (**i**) Left: confocal z-projected image of a control PV+ axon (red) immunostained with Syt2 (green). Right: surface-rendered 3D image of the same axon. (**j**) Example sections of 3D reconstructions. (**k**) Cuprizone increased the Syt2+ puncta spacing by approximately twofold (Mann–Whitney test ****p<0.0001, n = 12). Data shown as mean ± SEM and open circles individual axons or pairs. n.s., not significant.

The online version of this article includes the following figure supplement(s) for figure 3:

**Figure supplement 1.** Cuprizone decreases miniature inhibitory postsynaptic currents (mIPSCs) and somatodendritic parvalbumin (PV) puncta.

**Figure supplement 2.** Putative parvalbumin-positive (PV+) chandelier inputs at the axon initial segment (AIS) are unaffected by cuprizone-induced demyelination.

**Figure supplement 3.** Demyelination and dysmyelination reduces miniature inhibitory postsynaptic currents (IPSCs) and perisomatic Syt2+ puncta.

**Figure supplement 4.** Demyelination does not affect excitatory drive of parvalbumin-positive (PV+) basket cells (BCs).

The uIPSC failures and impairment of temporary facilitation may reflect failure of AP propagation along demyelinated PV axons, changes in the GABA release probability or a lower number of active release sites (<5, *Tamás et al., 1997*; *Thomson et al., 1996*). To further examine the properties of inhibition at L5 PNs, we recorded miniature inhibitory postsynaptic currents (mIPSCs). In line with the uIPSC findings, the results showed that mIPSCs were significantly reduced in peak amplitude (from ~20 to ~7 pA, p=0.002) without a change in frequency (*Figure 3—figure supplement 1c*). Furthermore, using PV immunofluorescence staining with biocytin-filled L5 PNs the number of $PV^+$ puncta was 40% reduced both at the soma and the primary apical dendrite, correlating with the overall reduction in immunofluorescent signals in cuprizone treatment (*Figure 3—figure supplement 1g–i*). Interestingly, in contrast to the loss of perisomatic $PV^+$ BC puncta, putative $PV^+$ chandelier cell inputs, identified by co-staining with the AIS marker ßIV-spectrin, were preserved (~8 puncta/AIS, Mann–Whitney test p=0.96, *Figure 3—figure supplement 2*). Furthermore, staining for Syt2, a $Ca^{2+}$ sensor protein selective for $PV^+$ presynaptic terminals (*Sommeijer and Levelt, 2012*; *Xu et al., 2007*) confirmed an ~35% reduction (*Figure 3—figure supplement 3a and b*). Together with the reduction in uIPSC peak amplitudes (*Figure 3f*), these data suggest that cuprizone-induced demyelination causes a loss of presynaptic $PV^+$ terminal sites. Interestingly, $Syt2^+$ puncta analysis in the dysmyelinated shiverer mouse line also showed a reduced number of $Syt2^+$ puncta at L5 PN somata and a reduced frequency of mIPSCs (p=0.019, *Figure 3—figure supplement 3c–g*), indicating that compact myelin is not only required for maintaining $PV^+$ interneuron inputs but also for $PV^+$ BC presynaptic terminal development.

Cuprizone treatment did not affect $PV^+$ axon length (*Figure 2g*, *Figure 2—figure supplement 1*), suggesting that the density of presynaptic terminals should be reduced. To test this idea, we performed Syt2 immunolabeling of individually biocytin-filled $PV^+$ BCs (*Figure 3i*). Consistent with the hypothesis, cuprizone treatment significantly reduced the density of $Syt2^+$ puncta by twofold (cuprizone, ~1 $Syt2^+$ puncta per 10 µm vs. 1 $Syt2^+$ puncta per 5 µm in control, Mann–Whitney test p<0.0001, *Figure 3j and k*). Interestingly, recordings of miniature excitatory postsynaptic currents (mEPSCs) from $PV^+$ interneurons of control and cuprizone-treated mice showed no changes in peak amplitude nor frequency (*Figure 3—figure supplement 4*), in keeping with the preservation of excitatory inputs onto L5 PNs following cuprizone-induced demyelination (*Hamada and Kole, 2015*) and suggesting that myelin loss has a significant impact on inhibitory synapse stabilization and maintenance.

Thus, myelin loss reduces the number of presynaptic sites, causing an increase of GABA release failures and a frequency-dependent depression, ultimately limiting the fast component of BC to PN inhibitory transmission.

## $PV^+$ activation rescues interictal spikes and theta oscillations, but not the loss of gamma

To understand how myelin deficits and loss of fast PV-mediated inhibition of PNs impacts network dynamics, we used AAV1-mediated delivery of Cre-dependent channelrhodopsin-2 (ChR2) into L5 of PV-Cre; Ai14 mice (*Figure 4a*). The ChR2 transduction rate was comparable between control and cuprizone mice (~70%, *Figure 4b and c*). In acute slices, we voltage-clamped L5 PNs and optogenetically evoked IPSC (oIPSC) with full-field blue light illumination (*Figure 4d*). Consistent with S1 L5 PNs receiving converging input from >100 $PV^+$ interneurons (*Packer and Yuste, 2011*), control oIPSCs rapidly facilitated to a peak amplitude of ~700 pA followed by rapid synaptic depression (*Figure 4f and g*). In slices from cuprizone mice, however, the oIPSC peak amplitude was significantly reduced (approximately twofold) while neither the steady-state amplitude during vesicle replenishment nor the total charge transfer reached a significant difference (control, –99.58 ± 28.5 pC vs. cuprizone, –54.3 ± 20.57 pC, p=0.236, n = 9 control and n = 8 cuprizone neurons, *Figure 4f and h*).

Impaired phasic $PV^+$ interneuron-mediated inhibition predicts a disrupted γ-rhythm. Experimental and computational studies show that in most cortical areas γ-rhythms are strongly shaped by electrically and synaptically coupled $PV^+$ interneurons, which, by temporally synchronizing firing rates, synaptic inhibitory time constants (≅ 9 ms), and the recurrent excitatory feedback from PNs, give rise to network resonance in the 30–80 Hz bandwidth (*Bartos et al., 2002*; *Cardin et al., 2009*; *Sohal et al., 2009*; *Traub et al., 1997*; *Wang and Buzsáki, 1996*). To test the cellular and circuit properties of the γ-rhythm, we examined the extent of evoked γ-modulation by leveraging optogenetic activation of $PV^+$ interneurons with AAV1-hChR2-YFP and introducing a laser fiber into L5 and recording the

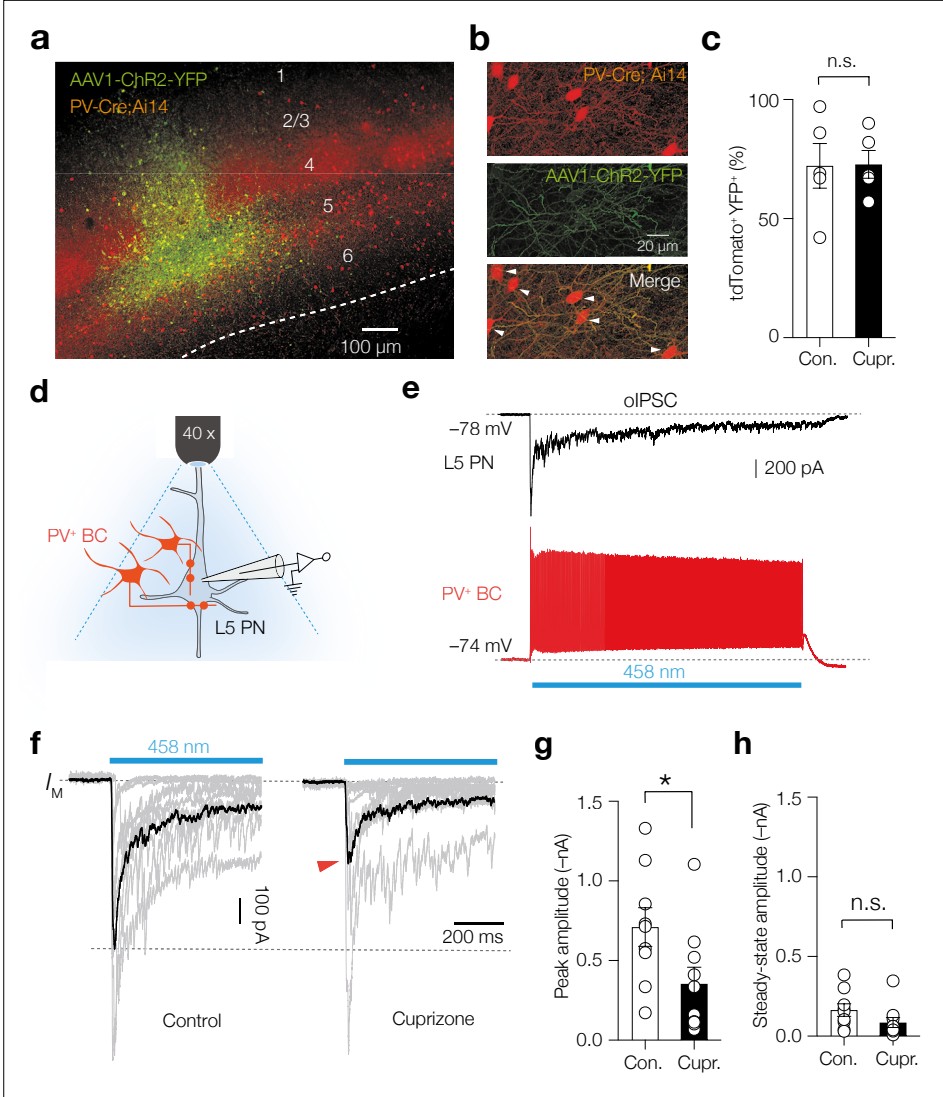

**Figure 4.** Demyelination impairs phasic parvalbumin-positive (PV⁺) basket cell (BC) inhibition of layer 5 (L5) pyramidal neurons (PNs). (**a**) Immunofluorescent image of AAV1-hChR2-YFP expression (green) injected into L5 of S1. (**b**) Confocal images of separate fluorescent channels showing td-Tomato⁺ cell bodies and neurites (red, top), the localization of AAV1-hChR2-YFP (green, middle), and the merge image (bottom). The majority of tdTomato⁺ cells were YFP⁺ (white arrows). (**c**) Average transfection rate of AAV1-hChR2-YFP in the L5 (>70%) is comparable in control and cuprizone conditions (Mann–Whitney test p=0.889, n = 5). (**d**) Schematic showing full-field blue light optogenetically evoked postsynaptic inhibitory currents (oIPSCs) in L5 PNs. (**e**) Example trace of a whole-cell current-clamp recording from a PV⁺ interneurons (bottom red) compared to a separate whole-cell voltage-clamp recordings from an L5 PN. A 1 s blue light field illumination (blue bar) produces sustained firing in *PV-Cre AAV1-ChR2* interneurons. (**f**) Single trial oIPSCs (gray) from different experiments (1 s duration pulses) overlaid with the average oIPSC (black) revealing a lower peak amplitude in cuprizone (red arrow). (**g**) Population data revealed an approximately twofold reduction in oIPSCs' peak amplitude (Mann–Whitney test p=0.0172). (**h**) Steady-state oIPSCs' amplitude did not reach significance, n.s., Mann–Whitney test p=0.0789.

LFP (*Figure 5a*). Evoking brief pulses of blue light (1 ms at a low gamma frequency of 30 Hz) showed that local circuit currents were modulated and highly phase-locked in slices from control mice (band-pass filter 25 and 40 Hz, *Figure 5b and c*, *Figure 5—figure supplement 1*). In striking contrast, no modulation or entrainment was observed in cuprizone-treated mice, neither when using high laser power (up to 6.5 mW, *Figure 5b–e*, *Figure 5—figure supplement 1*).

Could the diminished PV⁺ BC activity cause the emergence of θ rhythm and interictal spikes during quiet behavioral states of wakefulness? To test the direct contribution of PV⁺ BCs, we activated ChR2

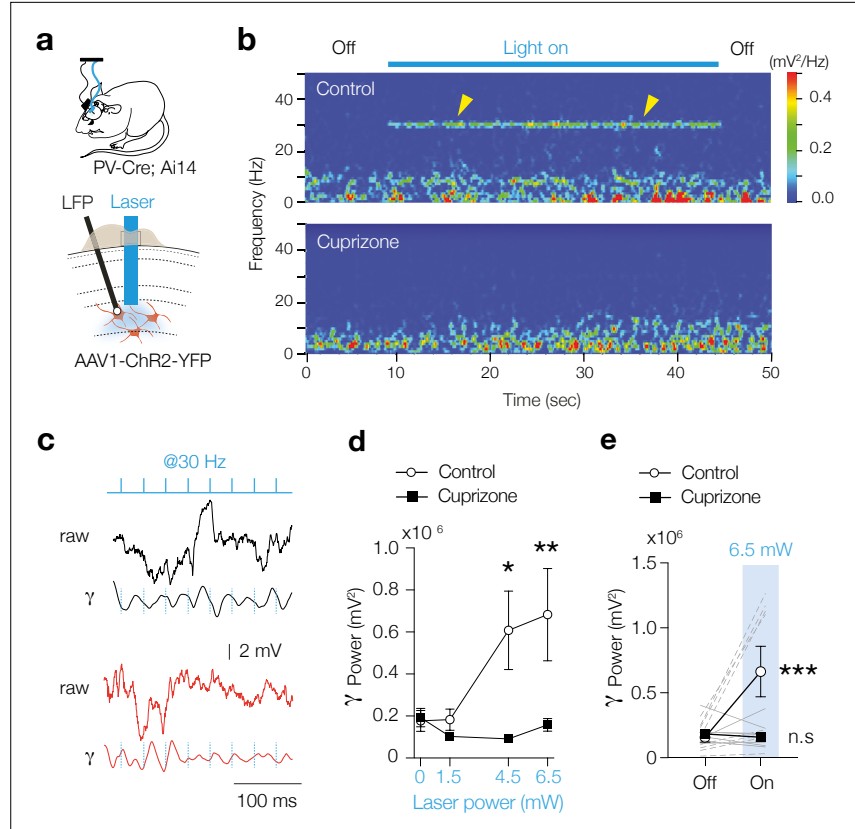

**Figure 5.** Demyelination impairs optogenetically evoked γ entrainment. (**a**) Schematic for chronic local field potential (LFP) recordings and in vivo optogenetic stimulation in freely moving PV-Cre; Ai14 mice. (**b**) Time frequency plot showing low gamma frequency (γ) entrainment (40 blue light pulses of 1 ms duration at 30 Hz) in control but not in cuprizone mice. (**c**) Raw LFP (top) and bandpass-filtered trace (25–40 Hz, bottom) from control and cuprizone during low-γ entrainment. (**i**) Population data of γ power with increasing laser power reveals impaired γ in cuprizone-treated mice. (**d**) Myelin-deficient mice low-γ band entrainment to optogenetic stimuli (two-way ANOVA followed by Šidák's multiple comparisons cuprizone vs. control, 0 mW, p>0.999; 1.5 mW, p=0.979; 4.5 mW, *p=0.0221, 6.5 mW, **p=0.0046). (**e**) Data are shown as mean ± SEM with gray lines individual cells, gray lines individual mice. n.s., not significant.

The online version of this article includes the following figure supplement(s) for figure 5:

**Figure supplement 1.** Myelin loss abolishes optogenetically evoked entrainment of γ rhythm.

for 1 s duration pulses in PV-Cre; Ai14 mice to generate tonic GABA release (**Figure 6a**). In cuprizone-treated mice, we found that optically driving $PV^+$ interneurons normalized the LFP power in the θ band to control levels, without affecting δ, β, and γ rhythms (two-way ANOVA treatment × light p=0.0124, Šidák's multiple comparisons tests in cuprizone, light on vs. off; for δ, p=0.9975; θ, p=0.0076; β, p=0.9481; γ, p=0.9998, **Figure 6a–c**, **Source data 1**). Furthermore, activation of blue light significantly reduced the frequency of interictal epileptic discharge frequency (p=0.0089, **Figure 6d and e**, **Figure 6—video 1**). The normalization of cortical rhythms by elevating sustained $PV^+$-mediated activity suggests that $GABA_A$ receptors are insufficiently activated in the demyelinated cortex. Finally, to directly examine the role of $GABA_A$ receptors agonism in dampening global interictal spikes we administered a nonsedative dose of diazepam (2 mg/kg i.p.), an allosteric modulator of postsynaptic $GABA_A$ receptors, in cuprizone-treated mice (7-week treatment). The results showed that diazepam significantly suppressed the interictal epileptiform discharges in cuprizone mice, indicating a prominent role of GABA in the deficits of circuit excitability (**Figure 6f**, **Figure 6—figure supplement 1**).

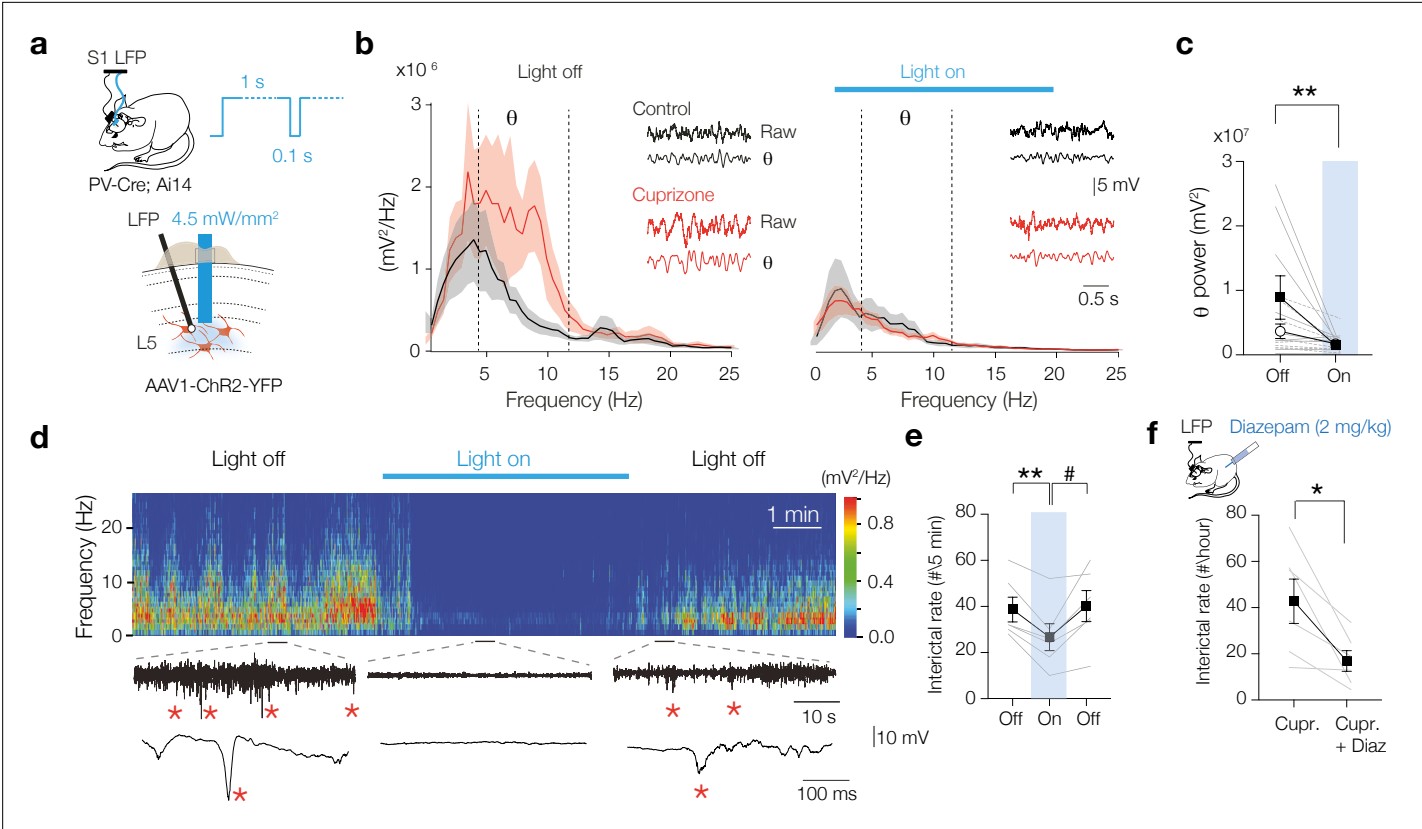

**Figure 6.** Optogenetic activation of myelin-deficient parvalbumin-positive (PV⁺) interneurons rescues theta rhythm and interictal epileptiform discharges. (**a**) Schematic drawing for chronic local field potential (LFP) and optogenetic stimulation in freely moving mice. A 1 s blue light pulse with 100 ms off periods activated PV⁺ interneurons. Blue light was switched on during high interictal activity (>10 spikes/min). (**b**) Power spectral content collected from 2 s epoch windows in control (black) and cuprizone (red) before (left) and during 3 min (right) optogenetic activation of PV⁺ interneurons. Insets: raw LFP signals (top) and $\theta$ content (bottom) in control (black) and cuprizone (red) condition. (**c**) Population data showing optogenetic activation of PV⁺ cells attenuated the amplified $\theta$ frequency in cuprizone mice to control levels (Šidák's multiple comparisons test in cuprizone, light on vs. light off for $\theta$, **p=0.0076). (**d**) Example time frequency plot (top) and raw LFP traces (below) showing suppression of interictal spikes during light on conditions. See also *Figure 6—video 1*. (**e**) Population data of transient optogenetic suppression of the interictal activity in 8-week cuprizone-treated mice (one-way ANOVA p=0.0165, Tukey's multiple comparisons tests; off vs. on, **p=0.0089; on vs. off, p=0.0539; off vs. off, p=0.928). (**f**) 2 mg/kg i.p. injection of diazepam in cuprizone-treated mice reduces interictal rate for at least 10 hr (two-tailed Student's *t*-test *p=0.024). Data shown as mean ± SEM and gray lines individual mice.

The online version of this article includes the following video and figure supplement(s) for figure 6:

**Figure supplement 1.** GABA_A receptor agonism suppresses interictal epileptiform discharge frequency.

**Figure 6—video 1.** Optogenetic activation of myelin-deficient parvalbumin-positive (PV⁺) interneurons attenuates interictal spikes.
https://elifesciences.org/articles/73827/figures#fig6video1

## Discussion

In this study, we identified that the cellular microarchitecture of myelination of PV⁺ BCs is required for stimulus-induced fast gamma frequencies, limiting the power of slow cortical oscillations and interictal spikes during quiet wakefulness. Interictal discharges identified as spikes on the EEG are an important diagnostic criterium in epilepsy and reflect hypersynchronized burst firing of PNs and interneurons (*Cohen et al., 2002*; *Tóth et al., 2018*; *Zhou et al., 2007*). The brief episodic and generalized nature of ECoG spikes we recorded in both demyelinated and dysmyelinated cortex (~50–500 ms in duration and ~1/min) resemble interictal spikes reported in epilepsy models (*Kleen et al., 2010*; *Zhou et al., 2007*) and are concordant with recordings in the hippocampus of cuprizone-treated mice by *Hoffmann et al., 2008*. Here, we fundamentally extend the insights into interictal spikes by showing their spatiotemporal synchronization across cortical areas and hemispheres and a selective manifestation during the vigilance state of quiet wakefulness. During brain states of quiescence, for example,

when whiskers are not moving, whole-cell in vivo recordings in the barrel cortex reveal low-frequency (<10 Hz) highly synchronized membrane potential fluctuations of PNs and interneurons, during which fast-spiking PV$^+$ interneurons are dominating action potential firing (*Gentet et al., 2010*; *Poulet and Petersen, 2008*). Consistent with the state-dependent increased activity of cortical interneuron firing, we found a selective amplification of the LFP theta power during quiet wakefulness, which may be explained by the reduced intrinsic excitability of PV$^+$ BCs and deficiency of fast inhibitory transmission in the demyelinated cortex (*Figures 2 and 3*). In support of this conjecture, optogenetic inhibition of PV$^+$ interneurons in the normally myelinated cortex is sufficient to increase PN firing rates, elevating the power of slow oscillations and triggering epileptiform activity (*Brill et al., 2016*; *Veit et al., 2017*; *Yang et al., 2017*). While our optogenetic activation of demyelinated PV$^+$ BCs normalized the power in the theta bandwidth, these interneurons are not critical for generating theta (*Cardin et al., 2009*; *Sohal et al., 2009*). Physiological theta oscillations are strongly driven by the long-range corticothalamic circuitry and cortical PN firing during non-rapid eye movement (NREM), also called thalamocortical spindles (*Steriade, 1997*). Interestingly, recordings in epilepsy patients showed that interictal discharges occur frequently during NREM sleep stages and coupled with the spindle activity (*Dahal et al., 2019*; *Ujma et al., 2017*). Whether interictal spike-spindle coupling occurs in cuprizone-treated mice or specific sleep stages are affected is not known, but the present results warrant investigation of the corticothalamic loop in the generation of interictal discharges during demyelination.

A major limitation of the experimental toolbox available to experimentally study myelination is the lack of axon- or cell-type selectivity. Whether amplified theta- and abolished gamma-frequency oscillations are the consequence of PV$^+$ axon demyelination, the loss of excitatory axon myelination or the combination thereof, remains to be further examined when more refined genetic or molecular methods become available to interrogate oligodendroglial myelination of specific cell types. In the absence of such strategies, however, our in vivo experiments optogenetically driving selectively PV$^+$ interneurons or activating GABA$_A$ receptors (*Figure 5*) uncover important converging evidence for a role of interneurons in the amplification and synchronization of slow oscillations and epileptic discharges. Interestingly, while tonically driving action potential firing in myelin-deficient PV$^+$ interneurons with optogenetic activation was able to rescue theta hypersynchrony, using a 30 Hz stimulation regime failed to entrain the LFP in the low-gamma frequency. This may suggest that for gamma, precisely timed spike generation of PV$^+$ BCs alone is insufficient and may require in addition synchronized inhibitory synaptic transmission, which is selectively reduced in demyelinated PV$^+$ BCs. Alternatively, a specific circuit connectivity or presynaptic GABA release dynamics may be lost in cuprizone-treated mice. In future studies, the role of myelination could be further examined by exploring whether remyelination restores the ability of PV$^+$ interneurons to modulate gamma oscillations.

## Myelination of PV$^+$ axons determines synapse assembly and maintenance

The requirement of myelination of PV$^+$ BCs to generate gamma rhythms is surprising in view of its sparse distribution in patches of ~25 μm across <5% of the total axon length (*Micheva et al., 2021*; *Micheva et al., 2016*; *Stedehouder et al., 2019*; *Figure 2*, *Figure 2—figure supplement 1*). The sparseness of interneuron myelination previously raised the question whether myelin speeds conduction velocity in these axon types (*Micheva et al., 2016*; *Stedehouder et al., 2019*; *Stedehouder and Kushner, 2017*). In a genetic mouse model with aberrant myelin patterns along fast-spiking interneuron axons, the inferred conduction velocity was reduced (*Benamer et al., 2020*). In contrast, we found that the average uIPSC latency (~800 μs) in completely demyelinated axons was normal and well within the range of previous paired recordings between myelinated PV$^+$ BC and PNs (700–900 μs, *Miles, 1990*; *Rossignol et al., 2013*). Assuming a typical axonal path length of ~200 μm between the AIS and presynaptic terminals connected with a PN, combined with an ~250 μs delay for transmitter release, the calculated conduction velocity would be 0.4 m/s, consistent with optically recorded velocities in these axons (~0.5 m/s, *Casale et al., 2015*). Our paired recordings, made near physiological temperature (34–36°C), may have had a limited resolution to detect temporal differences and are not excluding changes in the order of microseconds. To study submillicecond changes, it may be necessary to employ simultaneous somatic and axonal whole-cell recording (*Hu and Jonas, 2014*) and/or high-resolution anatomical analysis of myelin along the axon path, which recently showed a small albeit positive correlation between percentage of myelination and conduction velocity (*Micheva*

*et al., 2021*). Furthermore, conduction velocity tuning by myelination of GABAergic axons may become more readily apparent for long-range projections. Another constraint of the present study is the lack of information on the nodes of Ranvier along demyelinated PV⁺ BC axons. Aberrant interneuron myelin development, causing myelination of branch points, impairs the formation of nodes of Ranvier (*Benamer et al., 2020*). Reorganization of nodal voltage-gated ion channel clustering also occurs with the loss of myelin or oligodendroglial-secreting factors causing deficits in action potential propagation (*Freeman et al., 2015*; *Lubetzki et al., 2020*). How PV⁺ BC interneuron myelin loss changes the nodal ion channel distribution remains to be examined.

Converging evidence from the two distinct models (shiverer and cuprizone) showed that interneuron myelination critically determines PV⁺ release site number, dynamics, and connection probability (*Figure 3*, *Figure 3—figure supplement 3*), concordant with the observed synapse loss in Purkinje axons of the *les* rat (*Barron et al., 2018*). The molecular mechanisms how compact myelination of proximal axonal segments establishes and maintains GABAergic terminals in the higher-order distal axon collaterals are not known but may relate to its role in supplying metabolites to the axon (*Fünfschilling et al., 2012*). The PV⁺ interneuron myelin sheath contains high levels of noncompact 2′,3′-cyclic nucleotide 3′-phosphodiesterase (CNP) protein (*Micheva et al., 2018*; *Micheva et al., 2016*), which is part of the inner cytoplasmic inner mesaxon (*Edgar et al., 2009*). In the absence of inner cytoplasmic loops of oligodendroglial myelin the interneuron axons may lack sufficient trophic support to maintain GABAergic presynaptic terminals. Another possibility is pruning of the presynaptic terminals by microglia (*Chen et al., 2014*; *Favuzzi et al., 2021*; *Ramaglia et al., 2021*). Microglia become increasingly activated during sub-demyelinating stages within the first week of cuprizone treatment (*Caprariello et al., 2018*; *Skripuletz et al., 2013*) and in aged *Mbp⁺/⁻* mice (*Poggi et al., 2016*). In future studies, it needs to be examined whether attenuation of microglia activation could protect against PV⁺ synapse loss and interictal epileptiform discharges.

## Implications for cognitive impairments in gray matter diseases

The identification of a cellular mechanism for interictal spikes may shed light on the role of PV⁺ axon myelination in cognitive impairments in MS (*Benedict et al., 2020*) and possibly other neurological disorders. In preclinical models of epilepsy and epilepsy patients, interictal spikes have been closely linked to disruptions of the normal physiological oscillatory dynamics such as ripples required to encode and retrieve memories (*Cohen et al., 2002*; *Henin et al., 2021*; *Kleen et al., 2013*; *Kleen et al., 2010*). Interictal epileptic discharges are also observed in other neurodegenerative diseases, including Alzheimer (*Lam et al., 2017*). Notably, reduced gray matter myelination and oligodendroglia disruption are reported in multiple epilepsy models and recently in Alzheimer (*Chen et al., 2021*; *Drenthen et al., 2020*). Therefore, the cellular and circuit functions controlled by PV⁺ interneurons may represent a common mechanism for memory impairments in neurological disease encompassing myelin pathology. In support of this idea, neuropathological studies in MS show a specific loss of PV⁺ interneuron synapses in both cortex and hippocampus (*Ramaglia et al., 2021*; *Zoupi et al., 2021*). In MS patients, increased connectivity and synchronization in delta and theta band rhythms during resting state or task-related behavior have been reported (*Schoonheim et al., 2013*; *Tewarie et al., 2014*) and low GABA levels in sensorimotor and hippocampal areas are correlated with impairments of information processing speed and memory (*Cawley et al., 2015*; *Gao et al., 2018*). Taken together with the present work, promoting PV⁺ interneuron myelination, and thereby strengthening fast inhibition, may provide important new therapeutic avenues to improve cognition.

## Materials and methods

### Key resources table

| Reagent type (species) or resource | Designation | Source or reference | Identifiers | Additional information |
|---|---|---|---|---|
| Strain, strain background (*Mus musculus*, male/female) | C57BL6 | Janvier Labs | Cat# 2670020; RRID:MGI:2670020 | |
| Strain, strain background (*M. musculus*, male/female) | B6;129P2-*Pvalb*^tm1(cre)Arbr^/J | Jackson Laboratory | Stock no: 008069; RRID:IMSR_JAX:008069 | |

*Continued on next page*

*Continued*

| Reagent type (species) or resource | Designation | Source or reference | Identifiers | Additional information |
|---|---|---|---|---|
| Strain, strain background (*M. musculus,* male/female) | B6;129S6-*Gt(ROSA)26Sor^tm14(CAG-tdTomato)Hze*/J | Jackson Laboratory | Stock no: 007908; RRID:IMSR_JAX:007908 | |
| Strain, strain background (*M. musculus,* male/female) | C3Fe.SWV-*Mbp^shi*/J | Jackson Laboratory | Stock no: 001428; RRID:IMSR_JAX:001428 | |
| Transfected construct (*M. musculus*) | pAAV-EF1a-double-floxed-hChR2(H134)-EYFP-WPRE-HGHpA | Addgene.org | #20298; RRID:Addgene_20298 | |
| Antibody | Rabbit monoclonal, anti-MBP | Santa Cruz | Cat# sc-13564; RRID:AB_675707 | (1:300) |
| Antibody | Mouse monoclonal, anti-PV | Swant | Cat# 235; RRID:AB_10000343 | (1:1000) |
| Antibody | Rabbit polyclonal, anti-syt2 | Synaptic Systems | Cat# 105223; RRID:AB_10894084 | (1:500) |
| Antibody | Rabbit polyclonal, anti-ßIV-spectrin | M.Ransband (BCM) | N/A | (1:200) |
| Peptide, recombinant protein | Alexa 488-streptavidin | Thermo Fisher | Cat# S32354; RRID:AB_2315383 | (1:500) |
| Peptide, recombinant protein | Alexa 633-streptavidin | Thermo Fisher | Cat# S21375; RRID:AB_2313500 | (1:500) |
| Chemical compound, drug | Bis(cyclohexanone) oxaldihydrazone | Merck | Cat# C9012 | |
| Chemical compound, drug | Diazepam | Centrafarm Nederland B.V | Cat# RVG56691 | |
| Chemical compound, drug | Biocytin | Sigma | Cat# B4261 | |
| Chemical compound, drug | D-AP5 | HelloBio | Cat# 0225 | |
| Chemical compound, drug | Gabazine | Sigma | Cat# S106 | |
| Chemical compound, drug | CNQX | HelloBio | Cat# HB0205 | |
| Chemical compound, drug | Tetrodotoxin (TTX) citrate | Tocris | Cat# 1069 | |
| Software, algorithm | Neurolucida 360 Neurolucida Explorer | MBF Bioscience | V2018.02; RRID:SCR_001775 | |
| Software, algorithm | Synaptic puncta quantification | FIJI (ImageJ) | V2.0, V.2.0.0-rc-65/1.5w; RRID:SCR_002285 | |
| Software, algorithm | Neuroarchiver tool | Open Source Instruments; *Dubey et al., 2018* | LWDAQ_8.5.29 | |
| Software, algorithm | Igor pro 8 | WaveMetrics | V8.04 RRID:SCR_000325 | |
| Software, algorithm | AxoGraph | AxoGraph | RRID:SCR_014284 | |
| Software, algorithm | GraphPad | Prism 8 and 9 | RRID:SCR_002798 | |
| Software, algorithm | SP8 X (DM6000 CFS) | Leica application Suite | AF v3.2.1.9702; RRID:SCR_013673 | |
| Other | Electrode (90% Pt, 10% Ir) | Science Products | Cat# 101R-5T | |
| Other | Optical fiber | Thorlabs | Cat# FP200URT | |
| Other | Ceramic ferrule | Thorlabs | Cat# CFLC230-10 | |
| Other | Fiber coupled blue laser | Shanghai Laser & Optics Co. | Cat# 473 nm, DPSS Laser with fiber coupled (T3) | |

*Continued on next page*

*Continued*

| Reagent type (species) or resource | Designation | Source or reference | Identifiers | Additional information |
|---|---|---|---|---|
| Other | Cyclops LED driver | Open-ephys.org | Cat# Cyclops LED driver 3.7 | |
| Other | Patch-clamp amplifier | Dagan Corporates | BVC-700A | |
| Other | Patch-clamp amplifier | Molecular device | Axon Axopatch 200B | |
| Other | In vivo Multi Channel Systems (Portable-ME-systems) amplifier | Multi Channel Systems | ME16-FAO-uPA | |

## Animals

We crossed *Pvalb^Cre* mice (B6;129P2-*Pvalb^tm1(cre)Arbr*/J, stock no: 008069, Jackson Laboratory, RRID:IMSR_JAX:008069) with the Ai14 Cre reporter line B6;129S6-*Gt(ROSA)26Sor^tm14(CAG-tdTomato)Hze*/J (stock no: 007908, Jackson Laboratory, RRID:IMSR_JAX:007908). For other experiments, we used C57BL/6 mice (Janvier Labs, Saint-Berthevin Cedex, France, RRID:MGI:2670020). Shiverer mice were obtained from Jackson (C3Fe.SWV-*Mbp^shi*/J, stock no: 001428, RRID:IMSR_JAX:001428) and back-crossed with C57BL/6 mice for >10 generations. All mice were kept on a 12:12 hr light-dark cycle (lights on at 07:00, lights off at 19:00) with ad libitum food and water. For cuprizone treatment, either PV-Cre; Ai14 or C57BL/6 male or female mice, from 7 to 9 weeks of age, were fed ad libitum with normal chow food (control group) or were provided 0.2% (w/w) cuprizone (Bis(cyclohexanone)oxaldi-hydrazone, C9012, Merck) added either to grinded powder food or to freshly prepared food pellets (cuprizone group). Cuprizone-containing food was freshly prepared during every second or third day for the entire duration of the treatment (6–9 weeks). The average maximum weight loss during cupri-zone feeding was ~11% (n = 31). All animal experiments were done in compliance with the European Communities Council Directive 2010/63/EU effective from 1 January 2013. The experimental design and ethics were evaluated and approved by the national committee of animal experiments (CCD, application number AVD 80100 2017 2426). The animal experimental protocols were designed to minimize suffering and approved and monitored by the animal welfare body (IvD, protocol numbers NIN17.21.04, NIN18.21.02, NIN18.21.05, NIN19.21.04, and NIN20.21.02) of the Royal Netherlands Academy of Arts and Science (KNAW).

## In vitro electrophysiology

Mice were briefly anesthetized with 3% isoflurane and decapitated or received a terminal dose of pentobarbital natrium (5 mg/kg) and were transcardially perfused with ice-cold artificial CSF (aCSF) of the composition (in mM): 125 NaCl, 3 KCl, 25 glucose, 25 NaHCO$_3$, 1.25 Na$_2$H$_2$PO$_4$, 1 CaCl$_2$, 6 MgCl$_2$, 1 kynurenic acid, saturated with 95% O$_2$ and 5% CO$_2$, pH 7.4. After decapitation, the brain was quickly removed from the skull and parasagittal sections (300 or 400 µm) containing the S1 cut in ice-cold aCSF (as above) using a vibratome (1200S, Leica Microsystems). After a recovery period for 30 min at 35°C, brain slices were stored at room temperature. For patch-clamp recordings, slices were transferred to an upright microscope (BX51WI, Olympus Nederland) equipped with oblique illu-mination optics (WI-OBCD; numerical aperture, 0.8). The microscope bath was perfused with oxygen-ated (95% O$_2$, 5% CO$_2$) aCSF consisting of the following (in mM): 125 NaCl, 3 KCl, 25 D-glucose, 25 NaHCO$_3$, 1.25 Na$_2$H$_2$PO$_4$, 2 CaCl$_2$, and 1 MgCl$_2$. L5 PNs were identified by their typical large triangular shape in the infragranular layers and in slices from PV-Cre; Ai14 mice the PV$^+$ interneurons expressing tdTomato were identified using X-Cite series 120Q (Excelitas) with a bandpass filter (exci-tation maximum 554 nm, emission maximum 581 nm). Somatic whole-cell current-clamp recordings were made with a bridge current-clamp amplifier (BVC-700A, Dagan Corporation, USA) using patch pipettes (4–6 MΩ) filled with a solution containing (in mM): 130 K-gluconate, 10 KCl, 4 Mg-ATP, 0.3 Na$_2$-GTP, 10 HEPES, and 10 Na$_2$-phosphocreatine, pH 7.4, adjusted with KOH, 280 mOsmol/kg, to which 10 mg/mL biocytin was added. Voltage was analog low-pass filtered at 10 kHz (Bessel) and digitally sampled at 50–100 kHz using an analog-to-digital converter (ITC-18, HEKA Electronic) and data acquisition software AxoGraph X (v.1.7.2, AxoGraph Scientific, RRID:SCR_014284). The access resistance was typically <20 MΩ and fully compensated for bridge balance and pipette capacitance. All reported membrane potentials were corrected for experimentally determined junction potential

of –14 mV. Analysis for the electrophysiological properties includes PV$^+$ interneuron recordings from cells in normal ACSF and in the presence of CNQX and d-AP5 with high chloride intracellular solution (see below).

## mIPSC and mEPSC recordings

Whole-cell voltage-clamp recordings were made with an Axopatch 200B amplifier (Molecular Devices). Patch pipettes with a tip resistance of 3–5 MΩ were pulled from thins wall borosilicate glass. During recording, a holding potential of –74 mV was used. Both the slow- and fast pipette capacitance compensation were applied, and series resistance compensated to ~80–90%. Patch pipettes were filled with high chloride solution containing (in mM) 70 K-gluconate, 70 KCl, 0.5 EGTA, 10 HEPES, 4 MgATP, 4 K-phosphocreatine, 0.4 GTP, pH 7.3 adjusted with KOH, 285 mOsmol/kg and IPSCs isolated by the presence of the glutamate receptor blockers 6-cyano-7-nitroquinoxaline-2,3-dione (CNQX, 20 µM), d-2-amino-5-phosphonovaleric acid (d-AP5, 50 µM) and the sodium (Na$^+$) channel blocker tetrodotoxin (TTX, 1 µM Tocris). Individual traces (5 s duration) were filtered with a high-pass filter of 0.2 Hz and decimated in AxoGraph software (RRID:SCR_014284). Chart recordings of mIPSCs were analyzed with a representative 30 ms IPSC template using the automatic event detection tool of AxoGraph. Detected events were aligned and averaged for further analysis of inter-event intervals (frequency) and peak amplitude. For mEPSC recordings from PV$^+$ interneurons, we filled patch pipettes with a solution containing (in mM) 130 K-gluconate, 10 KCl, 4 Mg- ATP, 0.3 Na$_2$-GTP, 10 HEPES, and 10 Na$_2$-phosphocreatine, pH 7.4, adjusted with KOH, 280 mOsmol/kg and both gabazine (4 µM) and TTX (1 µM) were added to the bath solution. The mEPSCs were analyzed using events detection tool in AxoGraph. The recorded signals were bandpass filter (0.1 Hz to 1 kHz) and recordings analyzed with a representative 30 ms EPSC template, after which selected EPSCs aligned and averaged for further analysis of inter-event intervals (frequency) and peak amplitude.

## uIPSC recording and analysis

PV$^+$ interneurons (visually identified in PV-Cre; Ai14 mice based on tdTomato fluorescence expression) were targeted for whole-cell current-clamp recording within a radius of 50 µm from the edge of the L5 soma recorded in voltage-clamp configuration. APs in PV$^+$ interneurons were evoked with a brief current injection (1–3 ms duration) and uIPSCs recorded in the L5 PN from a holding potential of –74 mV. Only responses with 2× S.D. of baseline noise were considered being connected. Both fast and slow capacitances were fully compensated, series resistance compensation was applied to ~80–90%, and the current and voltage traces acquired at 50 kHz. For stable recordings with >50 uIPSCs, the episodes were temporally aligned to the AP and the uIPSCs were fit with a multiexponential function in Igor Pro. The curve fitting detected the baseline, uIPSC onset, rise time, peak amplitude, and decay time and was manually monitored. Fits were either accepted or rejected (e.g., when artifacts were present) and the number of uIPSC failures was noted for each recording.

## In vitro optogenetics

50 nL of AAV1 particles (titer $1 \times 10^{12}$ cfu/mL) produced from pAAV-EF1a-double-floxed-hChR2(H134)-EYFP-WPRE-HGHpA (Addgene.org #20298, RRID:Addgene_20298) was injected into L5 of S1 (coordinates from bregma; AP 0.15 mm, ML 0.30 mm, and DL 0.75 mm) of 6–9-week-old PV-Cre; Ai14 mice. About 7 days after the injection, a subset of mice was placed on 0.2% cuprizone diet for 8–9 weeks. PV$^+$ interneurons expressing hChR2 were identified using td-tom and YFP co-expression. Whole-cell voltage-clamp recordings were made from L5 PNs and oIPSCs were evoked with a X-cite 120Q, fluorescent lamp using filter BA460-510 (Olympus) in the presence of CNQX (50 µM) and dAP5 (20 µM) in the bath solution. The oIPSCs were evoked by illumination of large field with five light pulses of each 1 ms and 100 ms apart. Peak amplitude and area under the curve (charge) of oIPSC were quantified using AxoGraph. Only the first pulse was used for the quantification.

## In vivo electrophysiology and automated event detection

Chronic ECoG and LFP recordings were performed using in-house-made electrodes of platinum-iridium wire (101R-5T, 90% Pt, 10% Ir, complete diameter of 200 µm with 127 µm metal diameter, Science Products). The perfluoroalkoxy alkanes (PFA)-coated wire platinum-iridium wire was only exposed at the tip to record the LFP. For placement of the recording electrode, animals were anesthetized with

isoflurane (3%, flow rate 0.8 L/min with maintenance 1.5–1.8%, flow rate 0.6 L/min). A 1 cm midline sagittal incision was made starting above the interaural line and extending along the neck to create a pocket for subcutaneous placement of the transmitter along the dorsal flank of the animal. The recording electrodes in each hemisphere (stereotaxic coordinates relative to bregma: S1; –0.15 mm anterior and ±0.30 mm lateral; for LFP; ventral 0.75 mm, V1; 0.40 mm anterior and ±0.30 mm lateral; for LFP; ventral 0.75 mm) and ground electrode (6 mm posterior and 1 mm lateral) were implanted subdurally through small holes drilled in the skull, held in place with stainless steel screws (A2-70, Jeveka), and subsequently sealed with dental cement. Mice were provided with Metachem analgesic (0.1 mg per kg) after surgery and allowed to recover for 4–7 days before recordings. To obtain multiple hours recordings of ECoG-LFP at multiple weeks, mice remained in their home cage during an overnight recording session. ECoG-LFP data were collected using a ME2100-system (Multi Channel Systems); ECoG-LFP data were acquired at a sampling rate of 2 kHz using the multi-channel experimenter software (Multi Channel Systems). An additional 0.1–200 Hz digital bandpass filter was applied before data analysis. Large noise signals, due to excessive locomotion or grooming, were manually removed from the data. The ECoG and LFP recordings were processed offline with the Neuroarchiver tool (Open Source Instruments, http://www.opensourceinstruments.com/Electronics/A3018/Seizure_Detection.html). To detect interictal spikes, an event detection library was built as described previously (**Dubey et al., 2018**). During the initial learning phase of the library, the observer, if needed, overruled the identity of each new event by the algorithm until automated detection reached a false positive rate <1%. Subsequently, the ECoG-LFP data were detected by using a single library across all ECoG-LFP recordings. For determining the interictal rate, only S1 LFP signals were used for quantification.

## In vivo optogenetics with simultaneous ECoG-LFP recordings

50 nL of AAV1 particles (titer $1 \times 10^{12}$ cfu/mL) produced from pAAV-EF1a-double-floxed-hChR2(H134)-EYFP-WPRE-HGHpA (Addgene #20298, RRID:Addgene_20298) was injected unilaterally into the L5 of S1 (coordinates from bregma; AP 0.15 mm, ML 0.30 mm, and DL 0.75 mm) of 6–9-week-old PV-Cre; Ai14 mice. ECoG-LFP electrode (stereotaxic coordinates relative to bregma: –0.15 mm anterior and ±0.30 mm lateral; for LFP; ventral 0.75 mm) and ground electrode (6 mm posterior and 1 mm lateral) were implanted through small holes drilled in the skull, held in place with stainless steel screws (A2-70, Jeveka). Through the drilled hole, a polished multimode optical fiber (FP200URT, Thorlabs) held in ceramic ferrule (CFLC230-10, Thorlabs) was driven into layer 5 and ~50 μm above virus injection site. Once optical fiber and electrode were correctly placed, the drilled hole subsequently sealed with dental cement. A blue fiber-coupled laser (473 nm, DPSS Laser T3, Shanghai Laser & Optics Co.) was used to activate the ChR2. Cyclops LED Driver (Open Ephys), together with customized program, was used to design the on and off state of the laser. The driving signal from LED driver was also recorded at one of the empty channels in multichannel systems. This signal was used to estimate the blue light on or off condition. For gamma entrainment in S1, 40 pulses of blue light were flashed with 1 ms on and 28 ms off pulse.

To inhibit interictal spikes, 300 pulses of blue light were flashed with 1 s on and 100 ms off by manual activation of light pulses when periods of high interictal spikes were observed (>10 interictals/min). Aged-matched control mice were stimulated during the resting phase of the EEG, which was estimated using online EMG signal and video observation. For interictal counts, 5 min LFP signals were used from before light stimulation, during, and post light stimulation. Interictals were detected using event detection library. For analysis of the cortical rhythms, epochs were extracted using 2 s window at the start and after 180 pulses of blue light. Epoch-containing interictals were not included in the analysis.

For pharmacology experiment, continuous LFP recordings of >10–12 hr duration from the circadian quiet phase (from 19:00 to 09:00) of six cuprizone mice (7-week treatment) and three control mice were used for the analysis. To activate $GABA_A$ receptors in cuprizone-treated mice, we used diazepam (Centrafarm Nederland B.V) prepared in a 10% solution of (2-hydroxypropyl)-β-cyclo-dextrin (Sigma-Aldrich). A nonsedative dose of 2 mg/kg diazepam was injected intraperitoneally, and data was acquired for a period of 10 hr, starting 15 min after injection of drug in control and cuprizone mice. The automated event detection library (*Figure 1—figure supplement 1*) was used to determine the event frequency before and after diazepam injection.

## In vivo power spectrum analysis

Power spectral density (PSD) analysis was done using multitaper PSD toolbox from Igor Pro 8.0 (RRID:SCR_000325). The absence of high-voltage activity in the EMG electrode was classified as quiet wakefulness (*Figure 1—figure supplement 1*, *Figure 1—video 1*). For PSD analysis during inter-ictal activity, a 2 s window was used to extract LFP signal epochs. Epochs from control animals were selected comparing the EMG activity with cuprizone EMG activity. The interictal activity itself was excluded from the analysis. Selected LFP epochs were bandpass filtered between different frequency bands; delta, δ (0.5–3 Hz), theta, θ (4–12 Hz), beta, β (12.5–25 Hz), and gamma, γ (30–80 Hz). Multitaper PSD function (Igor Pro 8.0) was applied to the filtered data to plot the power distribution within each frequency band. Area under the curve was measured for each frequency band to compare power density between the control and cuprizone groups.

## Immunohistochemistry

L5 PNs were filled with 10 mg/mL biocytin during whole-cell patch-clamp recording for at least 30 min. Slices were fixed for 30 min with 4% paraformaldehyde (PFA) and stored in 0.1 M phosphate buffered saline (PBS; pH 7.4) at 4°C. Fixed 400 µm slices were embedded in 20% gelatin (Sigma-Aldrich) and then sectioned with a Vibratome (VT1000 S, Leica Microsystems) at 80 µm. Sections were preincubated with blocking 0.1 M PBS containing 5% normal goat serum (NGS), 5% bovine serum albumin (BSA; Sigma-Aldrich), and 0.3% Triton-X (Sigma) during 2 hr at 4°C to make the membrane permeable. For biocytin-labeled cells, streptavidin biotin-binding protein (Streptavidin Alexa 488, 1:500, Invitrogen, RRID:AB_2315383) was diluted in 5% BSA with 5% NGS and 0.3% Triton-X overnight at 4°C. Sections including biocytin-filled cells were incubated again overnight at 4°C with primary antibody rabbit anti-ßIV-spectrin (1:200; gift from M.N. Rasband, Baylor College of Medicine), mouse anti-MBP (1:250; Covance), mouse anti-PV (1:1000; Swant, RRID:AB_10000343), rabbit anti-syt2 (1:500, Synaptic Systems, RRID:AB_108 94084) in PBS blocking solution containing 5% BSA with 5% NGS and 0.3% Triton-X. Secondary antibody were used to visualize the immunoreactions: Alexa 488-conjugated goat anti-rabbit (1:500; Invitrogen), Alexa 488 goat anti-mouse (1: 500; Sanbio), Alexa 488 goat anti- guinea pig, Alexa 555 goat anti-mouse (1:500; Invitrogen), Alexa 555 goat anti-rabbit (1:500; Invitrogen), Alexa 633 goat anti-guinea pig (1:500; Invitrogen), Alexa 633 goat anti-mouse (1:500; Invitrogen), and Alexa 633 goat anti-rabbit (1:500; Invitrogen). Finally, sections were mounted on glass slides and cover slipped with Vectashield H1000 fluorescent mounting medium (Vector Laboratories, Peterborough, UK) and sealed.

## Confocal imaging

A confocal laser-scanning microscope SP8 X (DM6000 CFS; acquisition software, Leica Application Suite AF v3.2.1.9702, RRID:SCR_013673) with a ×63 oil-immersion objective (1.3 NA) and with 1× digital zoom was used to collect images of the labeled L5 neurons and the abovementioned proteins. Alexa fluorescence was imaged using corresponding excitation wavelengths at 15 units of intensity and a z-step of 0.3 µm. Image analysis was performed with Fiji (ImageJ) graphic software (v.2.0.0-rc-65/1.5w, National Institutes of Health, RRID:SCR_002285).

## Synaptic puncta counting and image analysis

The intensity of PV+ or Syt2 immunostaining was measured with a z-axis profile, calculating the mean RGB value for each z-plane. When quantifying the axosomatic puncta, the soma was defined to extend into the apical dendrite maximally ~4 µm and a boundary was drawn around the maximum edges (ROI). For counting apical dendritic puncta, a 200 µm length of apical dendrite was selected as ROI. Linear immunofluorescent signals from ßIV-spectrin were identified as AIS and used as ROI. For all analyses, the RGB images were separated into single-color channels using the color deconvolution plugin in ImageJ. The single-color channel containing boutons signals was subjected to thresholding and particle filter of 0.5 µm. The threshold was saved and applied to all images in the same staining group. The boutons were selected by scanning through the 3D projection of ROI with 0.35 µm z-steps. Trained experimenters identified the boutons either by colocalization of the ROI and PV/Syt2 or direct contact of the two. The boutons were characterized as round spots with a minimal radius of 0.5 µm ranging to almost 2 µm. Three experimenters blinded to the identity of the experiment group

independently replicated the results. All image analyses were done in Fiji (ImageJ) graphic software (v.2.0.0-rc-65/1.5w, National Institutes of Health, RRID:SCR_002285).

## PV⁺ axon reconstruction and quantification

For immunolabeling of biocytin-filled PV⁺ interneuron, 400 µm electrophysiology slices were incubated overnight at 4°C in PFA. Slices were rinsed with PBS followed by staining using streptavidin 488 (1:300, Jackson) diluted in PBS containing 0.4% Triton-X and 2% normal horse serum (NHS; Gibco) overnight at 4°C. Confocal images of 400-µm-thick slices were taken (see 'Confocal imaging') and immediately after thoroughly rinsed with 0.1 M PB and 30% sucrose at 4°C overnight. Next, slices were sectioned into 40 µm thick and preserved in 0.1 M PB before staining. Sections were preincubated in PBS blocking buffer containing 0.5% Triton-X and 10% NHS during 1 hr at room temperature. Sections were stained with primary mouse anti-MBP (1:300, Santa Cruz, RRID:AB_675707), rat anti-syt2 (RRID:AB_10894084) in 0.4% Triton-X, and 2% NHS with PBS solution for 72 hr. Alexa 488-conjugated secondary antibodies (1:300, Invitrogen) were added in PBS containing 0.4% Triton-X and 2% NHS, posterior to washing steps with PBS. Then, sections were mounted on slides and cover slipped with Vectashield H1000 fluorescent mounting medium, sealed, and imaged. Biocytin-labeled PV⁺ neurons were imaged using upright Zeiss LSM 700 microscope (Carl Zeiss) with ×10 and ×63 oil-immersion objectives (0.45 NA and 1.4 NA, respectively) and 1× digital zoom with step size of 0.5 µm. Alexa 488 and Alexa 647 were imaged using 488 and 639 excitation wavelengths, respectively. The 10× image was taken to determine the exact location of biocytin-filled cells. Subsequently, axonal images were taken at ×63 magnification. Axons were analyzed as described previously (*Stedehouder et al., 2019*) and identified by their thin diameter, smoothness, obtuse branching processes, and occasionally by the presence of the axon bleb. Images were opened in Neurolucida 360 software (v2018.02, MBF Bioscience, RRID:SCR_001775) for reconstruction using the interactive user-guided trace with the Directional Kernels method. Axon and myelinated segments were analyzed using Neurolucida Explorer (MBF Bioscience, RRID:SCR_001775). Axonal segments were accepted as myelinated when at least one MBP-positive segment colocalized with streptavidin across the internode length.

## Statistics

All statistical tests were performed using Prism 8 or 9 (GraphPad Software, LLC, San Diego, CA, RRID:SCR_014284). For comparisons of two independent groups, we used two-tailed Mann–Whitney *U*-tests. For multiple group comparisons, data were initially assessed for normality and subsequently we either used ordinary one-way ANOVA followed by Tukey's multiple comparisons or two-way ANOVA with repeated measures followed by Šidák's multiple comparisons tests to correct for multiple comparisons. The level of significance was set to 0.05 for rejecting the null hypothesis. A detailed overview of the statistical analyses performed in this study, together with the numbers used for figures and statistical testing, is provided in *Source data 1*.

## Acknowledgements

The authors are indebted to Prof. Dr. Stefan Hallermann (University of Leipzig) for providing the uIPSC analysis script. We thank Ms. Anouk Meuwissen, Catherine Jenkins, Denise de Ronde, and Dr. Koen Kole (NIN-KNAW) for their support in part of the recordings and optogenetic experiments. Sharon I De Vries performed the electron microscopy. We thank Dr. Corette Wierenga (UU) and Dr. David Vandael (NIN-KNAW) for providing highly valuable comments on earlier versions of the manuscript and experimental work. This work was in part funded by the National Multiple Sclerosis Society RG-1602-07777 (MK), The Netherlands Research Council NWO Vici 865.17.003 (MK), The Netherlands Research Council NWO 013.18.002 (SAK), and European Research Area Network ERA-NET NEURON JTC2018-024 (SAK).

# Additional information

## Funding

| Funder | Grant reference number | Author |
|---|---|---|
| National Multiple Sclerosis Society | RG-1602-07777 | Maarten HP Kole |
| Nederlandse Organisatie voor Wetenschappelijk Onderzoek | Vici 865.17.003 | Maarten HP Kole |
| Nederlandse Organisatie voor Wetenschappelijk Onderzoek | 013.18.002 | Steven A Kushner |
| ERA-NET | JTC2018-024 | Steven A Kushner |

The funders had no role in study design, data collection and interpretation, or the decision to submit the work for publication.

## Author contributions

Mohit Dubey, Data curation, Formal analysis, Investigation, Methodology, Project administration, Resources, Software, Supervision, Validation, Visualization, Writing – review and editing; Maria Pascual-Garcia, Formal analysis, Investigation, Methodology, Writing – review and editing; Koke Helmes, Formal analysis, Investigation, Methodology; Dennis D Wever, Formal analysis, Investigation; Mustafa S Hamada, Data curation, Formal analysis, Investigation, Methodology, Supervision, Visualization; Steven A Kushner, Funding acquisition, Methodology, Resources, Supervision, Validation, Writing – review and editing; Maarten HP Kole, Conceptualization, Data curation, Formal analysis, Funding acquisition, Investigation, Methodology, Project administration, Resources, Supervision, Validation, Visualization, Writing – original draft, Writing – review and editing

## Author ORCIDs

Mohit Dubey 
Mustafa S Hamada 
Steven A Kushner 
Maarten HP Kole 

## Ethics

All animal experiments were done in compliance with the European Communities Council Directive 2010/63/EU effective from 1 January 2013. The experimental design and ethics were evaluated and approved by the national committee of animal experiments (CCD, application number AVD 80100 2017 2426). The specific experimental protocols involving animals were designed to minimize suffering and approved and monitored by the animal welfare body (IvD, protocol numbers; NIN17.21.04, NIN18.21.02, NIN18.21.05, NIN19.21.04 and, NIN20.21.02) of the Royal Netherlands Academy of Arts and Science (KNAW).

## Decision letter and Author response

Decision letter https://doi.org/10.7554/eLife.73827.sa1
Author response https://doi.org/10.7554/eLife.73827.sa2

# Additional files

## Supplementary files

- Transparent reporting form
- Source data 1. Data and statistical analyses of all figures.

## Data availability

Raw data for Figure 1d is accessible via Dryad (doi:https://doi.org/10.5061/dryad.pk0p2ngpk).

The following dataset was generated:

| Author(s) | Year | Dataset title | Dataset URL | Database and Identifier |
|---|---|---|---|---|
| Dubey M | 2022 | Multichannel ECoG and LFP from control and cuprizone mice | https://doi.org/10.5061/dryad.pk0p2ngpk | Dryad Digital Repository, 10.5061/dryad.pk0p2ngpk |

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
