## [Editor Report]

This study shows that demyelination results in reduced inhibition by decreasing the number of release sites for the inhibitory transmitter GABA, thus promoting epileptic activity. Importantly, the paper describes that demyelination causes an increase in theta oscillation power during quiet wakefulness and an impairment of optogenetically induced γ oscillations in the cortex. These results expand our understanding of the functions of myelin in gray matter and its clinical relevance to demyelinating disorders such as multiple sclerosis.

---

## [Decision Letter]

**Decision letter after peer review:**

Thank you for submitting your article "Myelin speeds cortical oscillations by consolidating phasic parvalbumin-mediated inhibition" for consideration by *eLife*. Your article has been reviewed by 2 peer reviewers, and the evaluation has been overseen by a Reviewing Editor and Laura Colgin as the Senior Editor. The reviewers have opted to remain anonymous.

Essential revisions:

The conclusions of this paper are mostly well supported by data, but the following points need to be clarified:

(1) For the reader to fully understand the effects on theta and γ rhythms, it would be useful to briefly summarize somewhere the circuitry that produces these oscillations.

(2) For clarity: state in the methods, and also early in the main text, that data are (I assume, from reading Table S1 legend) displayed as mean+/-s.e.m. Give exact p values everywhere instead of stars, and state whether correction for multiple comparisons has been made where appropriate.

(3) State the age of the animals at the start of the results (not just in the methods) because the effects of demyelination may be age dependent (cf the Benamer and Angulo (2020) paper).

(4) In Figure 1e-f, contrary to the text it appears that with cuprizone γ power is increased more than 2-fold in quite wakefulness, and that theta power increases 50% when moving. Please give exact p values for these comparisons (and for the theta comparison in quiet wake), and state whether they are corrected for multiple comparisons.

(5) Showing that Shiverer mice also have seizures and interictal discharges is not really a robust test of the concern that (line 102) "interictal epileptiform discharges are due to off-target effects of cuprizone treatment". The authors could provide information on what happened to the γ and theta power in the MBP- mice, and also state whether cuprizone affected EPSCs in pyramidal cells or interneurons.

(6) In Line 141 it is stated that normally there was 80% myelination along 83 mm of axons analysed. This contrasts dramatically with the statement of Benamer et al., (2020, Nat Comm, see penultimate para in column 2 of page 11) that only 2.5-4% of each axon is myelinated (cf your Figure 2E). Can you explain this apparent difference (e.g. do you mean to state that 80% cells had at least some myelin on their axon)?

(7) Is it possible to define whether the main effect on function is due to a presumed lower firing rate of FSIs in vivo (due to their increased capacitance) or due to their 3-fold weaker IPSCs?

(8) Does the change in axon topology reported by Benamer et al., (2020 Nat Comm) contribute to the decreased probability of FSI-SSC connections being detected (see line 218)?

(9) In Lines 650-657 it is not really clear how, in the Syt2 analysis of Figure 3 and Figure S4, it was determined whether each of the many boutons visible in control conditions was associated with the axon or soma being studied. Picking them "by hand" seems a bit vague. When you state "colocalization of the pyramidal cell and PV/Syt2" the results obtained will depend critically on the thresholding applied for each label. More details on this are needed so that the reader can implement an objective and identical procedure if needed.

(10) Line 261 and following. State the frequency of optical excitation used. How does the γ entrainment depend on the frequency used? Is 30Hz stimulation needed, or would tonic inhibition work as well?

(11) Line 365. "Interestingly, in contrast to dampening theta oscillations driving myelin-deficient PV+ interneurons at γ-frequencies did not entrain local field oscillations." Can the authors explain why not? Is this because, even if action potentials are generated, the decrease in Syt2+ release sites results in insufficent GABA release onto the pyramidal cells?

(12) In Figure 5A legend, does the statement "Blue light was switched on during high interictal activity (>10 spikes/min)." apply to all of the rest of the figure? If so, say so.

(13) Line 379: Shouldn't "hypermyelinated" read HYPOmyelinated"?

*Reviewer #1:*

The study tests the presumption that demyelination of inhibitory interneurons in the cerebral cortex impairs neuronal oscillations. This is an important hypothesis because the function of myelin to speed impulse conduction along long-distant projections in white matter seems irrelevant to short-distance communication by interneurons, and it has been suggested that grey matter myelin facilitates appropriate intercortical oscillations, notably γ oscillations that are strongly influenced by inhibitory interneurons.

The approach used is to induce demyelination by cuprizone administration and measure the effects on optogenetically-induced γ oscillations in vivo and patch clamp recording of IPSCs in brain slice. The results show demyelination is associated with increased interictal discharges detected by ECoG, which is indicative of hypersynchronized burst firing; increased theta oscillation power during quiet wakefulness, but not when mice were actively exploring; impaired inhibition without a measurable change in conduction velocity, a loss of PV+ BC presynaptic terminals, and impaired entrainment with activity induced by optogenetic stimulation at the γ frequency. The data provided on these points are compelling and evaluated by appropriate statistical tests.

The studies provide needed data on the effects of demyelination of interneurons, which expands understanding of the functions of myelin in grey matter and has clinical relevance to demyelinating disorders such as multiple sclerosis.

A concern is that the system-wide demyelination is induced, not demyelination restricted specifically to the inhibitory neurons, but these difficulties are acknowledged in the discussion.

*Reviewer #2:*

Cognitive processing is thought to depend on theta and γ rhythms that are generated in part by inhibitory interneurons, notably parvalbumin-expressing cells. These interneurons are unusual in that they are wrapped by myelin over the proximal part of their axonal tree, even though the normal role of myelin (to speed conduction of the nerve impulse) seems irrelevant here because none of the axonal branches are very long.

Previous work by the Angulo group showed that disrupting the normal development of this myelin led to a weakening of the inhibition that these cells generate in excitatory pyramidal cells and thus led to cognitive changes. The current paper disrupts myelin in older animals by a different method – cuprizone – and also finds that this results in reduced inhibition, in part by increasing the cell capacitance and thus making the cell harder to excite, and in part by decreasing the number of release sites for the inhibitory transmitter GABA. Importantly the paper then goes on to assess how this affects the theta and γ rhythms generated in the cortex, and to show that the reduction of inhibition promotes epileptic seizures. The mechanisms are assessed using histology and immunohistochemistry (to quantify myelination, axon structure and transmitter release sites), electrophysiology (to assess synapse strength and cell spiking behavior), and optogenetics (to control the spiking of the cells).

The conclusions of this paper are mostly well supported by data, but the following points need to be clarified.

(1) For the reader to fully understand the effects on theta and γ rhythms, it would be useful to briefly summarise somewhere the circuitry that produces these oscillations.

(2) For clarity: state in the methods, and also early in the main text, that data are (I assume, from reading Table S1 legend) displayed as mean+/-s.e.m. Give exact p values everywhere instead of stars, and state whether correction for multiple comparisons has been made where appropriate.

(3) State the age of the animals at the start of the results (not just in the methods) because the effects of demyelination may be age dependent (cf the Benamer and Angulo (2020) paper).

(4) In Figure 1e-f, contrary to the text it appears that with cuprizone γ power is increased more than 2-fold in quite wakefulness, and that theta power increases 50% when moving. Please give exact p values for these comparisons (and for the theta comparison in quiet wake), and state whether they are corrected for multiple comparisons.

(5) Showing that Shiverer mice also have seizures and interictal discharges is not really a robust test of the concern that (line 102) "interictal epileptiform discharges are due to off-target effects of cuprizone treatment". The authors could provide information on what happened to the γ and theta power in the MBP- mice, and also state whether cuprizone affected EPSCs in pyramidal cells or interneurons.

(6) In Line 141 it is stated that normally there was 80% myelination along 83 mm of axons analysed. This contrasts dramatically with the statement of Benamer et al., (2020, Nat Comm, see penultimate para in column 2 of page 11) that only 2.5-4% of each axon is myelinated (cf your Figure 2E). Can you explain this apparent difference (e.g. do you mean to state that 80% cells had at least some myelin on their axon)?

(7) Is it possible to define whether the main effect on function is due to a presumed lower firing rate of FSIs in vivo (due to their increased capacitance) or due to their 3-fold weaker IPSCs?

(8) Does the change in axon topology reported by Benamer et al., (2020 Nat Comm) contribute to the decreased probability of FSI-SSC connections being detected (see line 218)?

(9) In Lines 650-657 it is not really clear how, in the Syt2 analysis of Figure 3 and Figure S4, it was determined whether each of the many boutons visible in control conditions was associated with the axon or soma being studied. Picking them "by hand" seems a bit vague. When you state "colocalization of the pyramidal cell and PV/Syt2" the results obtained will depend critically on the thresholding applied for each label. More details on this are needed so that the reader can implement an objective and identical procedure if needed.

(10) Line 261 and following. State the frequency of optical excitation used. How does the γ entrainment depend on the frequency used? Is 30Hz stimulation needed, or would tonic inhibition work as well?

(11) Line 365. "Interestingly, in contrast to dampening theta oscillations driving myelin-deficient PV+ interneurons at γ-frequencies did not entrain local field oscillations." Can the authors explain why not? Is this because, even if action potentials are generated, the decrease in Syt2+ release sites results in insufficent GABA release onto the pyramidal cells?

(12) In Figure 5A legend, does the statement "Blue light was switched on during high interictal activity (>10 spikes/min)." apply to all of the rest of the figure? If so, say so.

(13) Line 379: Shouldn't "hypermyelinated" read HYPOmyelinated"?

---

## [Author Response]

Essential revisions:The conclusions of this paper are mostly well supported by data, but the following points need to be clarified:(1) For the reader to fully understand the effects on theta and γ rhythms, it would be useful to briefly summarize somewhere the circuitry that produces these oscillations.

We agree this is important information and we have updated the text in the revised version. Regarding γ rhythms we write (line 241): “Experimental and computational studies show that in most cortical areas γ rhythms are strongly shaped by electrically and synaptically coupled PV^+^ interneurons which by temporally synchronizing firing rates, the synaptic inhibitory time constants (≅ 9 ms) and the recurrent excitatory feedback from PNs, giving rise to network resonance in the 30 to 80 Hz bandwidth”. Regarding theta rhythms “Physiological theta oscillations are strongly driven by the long-range corticothalamic circuitry and cortical PN firing during non-rapid eye movement (NREM), also called thalamocortical spindles” (line 294)

(2) For clarity: state in the methods, and also early in the main text, that data are (I assume, from reading Table S1 legend) displayed as mean+/-s.e.m. Give exact p values everywhere instead of stars, and state whether correction for multiple comparisons has been made where appropriate.

We agree with the reviewer that clarity about the data and statistical analyses is essential. We included in our original submission a complete overview of all statistical tests and the exact P values for all our data analyses in Table S2. We always used multiple comparisons where required (i.e. when more than two groups were included in one analysis). In the revision, we have now provided a single Excel file that besides the statistical test details with exact values also shows the raw data (Source Data 1) and have added statistical test details in the Figure Legends or main text, providing the exact test values for the stars in the Figures.

(3) State the age of the animals at the start of the results (not just in the methods) because the effects of demyelination may be age dependent (cf the Benamer and Angulo (2020) paper).

For all our demyelination experiments we used mice that were 7–9 weeks old at the start of cuprizone treatment. We have now added this information also at the start of the Results (Line 76). In the article of Benamer et al., 2020 there is no demyelination experiment performed so we are unsure how to address this point.

(4) In Figure 1e-f, contrary to the text it appears that with cuprizone γ power is increased more than 2-fold in quite wakefulness, and that theta power increases 50% when moving. Please give exact p values for these comparisons (and for the theta comparison in quiet wake), and state whether they are corrected for multiple comparisons.

As mentioned above, we should have made it clearer that a complete overview of statistical comparisons was available in Table S2 (now Source Data 1). We do not find statistical support for the notion that theta power is higher during locomotion (dotted lines) or that γ is increased during cuprizone treatment in the wake state (continuous lines). The two-way ANOVA analysis revealed that during quiet wakefulness the power is overall higher in the cuprizone treatment (ANOVA treatment *P* < 0.0001) but not specific to a bandwidth (interaction *P* = 0.0821). Accordingly, post-hoc comparisons also revealed no differences between conditions. In the revied manuscript, the exact P values are provided in the figure legends. We Šidák’s or Tukey’s post post hoc tests to correct for multiple comparisons (see new Source Data 1).

(5) Showing that Shiverer mice also have seizures and interictal discharges is not really a robust test of the concern that (line 102) "interictal epileptiform discharges are due to off-target effects of cuprizone treatment". The authors could provide information on what happened to the γ and theta power in the MBP- mice, and also state whether cuprizone affected EPSCs in pyramidal cells or interneurons.

Thank you for raising these valuable points. Analyzing the power of distinct bandwidths in the Shiverer mice was unfortunately problematic since we did not have an EMG electrode installed to identify the behavioral state of the animals. As an alternative, we have now analyzed the frequency band power in periods just before and after interictal spikes, to enable a direct comparison with rhythms in cuprizone-treated mice. The new analysis shows that the power across the four frequency bands around interictal spikes was similar between the two groups (Two-way ANOVA Frequency bands x model *P* = 0.7875, *n* = 6 mice, Figure 1—figure supplement 2).

In the original manuscript, we showed that miniature EPSCs from PV interneurons were not different in amplitude or frequency (in the revised version Figure 3—figure supplement 4). The lack of difference in excitatory synaptic currents is in line with our previous EPSP recordings from hippocampal and neocortical PNs in the cuprizone model (Ramaglia et al., 2021; Hamada et al., 2015).

(6) In Line 141 it is stated that normally there was 80% myelination along 83 mm of axons analysed. This contrasts dramatically with the statement of Benamer et al., (2020, Nat Comm, see penultimate para in column 2 of page 11) that only 2.5-4% of each axon is myelinated (cf your Figure 2E). Can you explain this apparent difference (e.g. do you mean to state that 80% cells had at least some myelin on their axon)?

We thank the reviewer for raising this issue. We quantified the proportion of axonal myelination by examining large areas within layer 5 and visual selection of PV^+^ axons, possibly biasing our analysis to larger diameter axons which are typically myelinated (>0.4 µm, Stedehouder et al., 2019). These results also may include long-range PV axons within the neocortex. To the best of our knowledge, Benamer et al., based the 2.5–4% on their experimentally measured total myelinated length combined with a literature value (10 to 22 mm; Koelbl et al., 2019). Using our detailed and extensive morphological analysis of single biocytin-filled PV cells stained for MBP we determined the percentage of PV axon myelination to be on average 2.8 ± 1.2% (n = 4 axons). We have added this estimate to the manuscript (line 133).

(7) Is it possible to define whether the main effect on function is due to a presumed lower firing rate of FSIs in vivo (due to their increased capacitance) or due to their 3-fold weaker IPSCs?

The reviewer raises an excellent question. In comparison to the 2.0-fold reduction in the optical IPSC peak amplitude, the threshold for action potential firing (rheobase) was increased by 1.4-fold. Although both aspects will effectively limit inhibition, our optogenetic rescue experiment showed that when we extrinsically drove BC action potential firing, γ entrainment was entirely abolished (Figure 5). Although this shows the impact of a lack of fast inhibition, it does not inform us whether it is more important in comparison to the increased threshold to action potential generation under in vivo conditions.

(8) Does the change in axon topology reported by Benamer et al., (2020 Nat Comm) contribute to the decreased probability of FSI-SSC connections being detected (see line 218)?

The reviewer asks whether we can explain the results published by Benamer et al., regarding their finding of an absence of connectivity between fast spiking interneurons (FSI) and spiny stellate cells (SSCs). However, we did not collect data that might offer insight into their findings. Moreover, the study of Benamer et al., has a different experimental design than ours since their FS interneuron axons remained myelinated. In our experience, with a complete loss of myelin in adult mice, we did not find differences in the topology of identified PV axons determined over millimeters of reconstructed axons (Figure 2).

(9) In Lines 650-657 it is not really clear how, in the Syt2 analysis of Figure 3 and Figure S4, it was determined whether each of the many boutons visible in control conditions was associated with the axon or soma being studied. Picking them "by hand" seems a bit vague. When you state "colocalization of the pyramidal cell and PV/Syt2" the results obtained will depend critically on the thresholding applied for each label. More details on this are needed so that the reader can implement an objective and identical procedure if needed.

We agree this was insufficiently detailed. We have changed the method description (from line 609 and onwards):

“The intensity of PV^+^ or Syt2 immunostaining was measured with a z-axis profile, calculating the mean RGB value for each z-plane. […] Three experimenters blinded to the identity of the experiment group independently replicated the results. All image analysis was done in Fiji (ImageJ) graphic software (v.2.0.0-rc-65/1.5w, National Institutes of Health)”.

(10) Line 261 and following. State the frequency of optical excitation used. How does the γ entrainment depend on the frequency used? Is 30Hz stimulation needed, or would tonic inhibition work as well?

For acute slice recordings, we used 1-second-long pulses of blue light for field illumination. We have not tested different optical stimulation frequencies in vivo. However, Cardin et al., 2009, showed that the peak of entrainment of the LFP to γ occurs at 50 Hz, when testing 10 Hz steps between 10 and 100 Hz. We are not aware of experiments that have examined whether tonic inhibition would work as well, but this could of course be technically challenging given the risk of depolarization block with prolonged continuous optogenetic stimulation.

(11) Line 365. "Interestingly, in contrast to dampening theta oscillations driving myelin-deficient PV+ interneurons at γ-frequencies did not entrain local field oscillations." Can the authors explain why not? Is this because, even if action potentials are generated, the decrease in Syt2+ release sites results in insufficent GABA release onto the pyramidal cells?

Yes, we agree with this viewpoint. Because of losing presynaptic PV terminals, or lacking synchronization of release sites, the reduced fast component in PV-mediated inhibition should limit the capacity for γ entrainment. See also our answer to point 7 above.

(12) In Figure 5A legend, does the statement "Blue light was switched on during high interictal activity (>10 spikes/min)." apply to all of the rest of the figure? If so, say so.

Thank you for pointing this out. In figure 5A, the blue light was switched “on” only during high interictal activity (>10 spikes/min). For optical γ entrainment (Figure 4), we used 1-ms duration pulses of blue light, repeated forty times, switched randomly during states of awake locomotor activity.

(13) Line 379: Shouldn't "hypermyelinated" read HYPOmyelinated"?

We have changed the wording to “aberrant myelination” since there was neither evidence for less nor for more myelination of fast-spiking interneurons (see their Supplementary Figure 2 of Benamer et al., 2020).

Reviewer #1:The study tests the presumption that demyelination of inhibitory interneurons in the cerebral cortex impairs neuronal oscillations. This is an important hypothesis because the function of myelin to speed impulse conduction along long-distant projections in white matter seems irrelevant to short-distance communication by interneurons, and it has been suggested that grey matter myelin facilitates appropriate intercortical oscillations, notably γ oscillations that are strongly influenced by inhibitory interneurons.The approach used is to induce demyelination by cuprizone administration and measure the effects on optogenetically-induced γ oscillations in vivo and patch clamp recording of IPSCs in brain slice. The results show demyelination is associated with increased interictal discharges detected by ECoG, which is indicative of hypersynchronized burst firing; increased theta oscillation power during quiet wakefulness, but not when mice were actively exploring; impaired inhibition without a measurable change in conduction velocity, a loss of PV+ BC presynaptic terminals, and impaired entrainment with activity induced by optogenetic stimulation at the γ frequency. The data provided on these points are compelling and evaluated by appropriate statistical tests.The studies provide needed data on the effects of demyelination of interneurons, which expands understanding of the functions of myelin in grey matter and has clinical relevance to demyelinating disorders such as multiple sclerosis.A concern is that the system-wide demyelination is induced, not demyelination restricted specifically to the inhibitory neurons, but these difficulties are acknowledged in the discussion.

We thank the reviewer for the positive evaluation of the manuscript and fully agree that there is a need for cell-type specific demyelination methods.

Reviewer #2:Cognitive processing is thought to depend on theta and γ rhythms that are generated in part by inhibitory interneurons, notably parvalbumin-expressing cells. These interneurons are unusual in that they are wrapped by myelin over the proximal part of their axonal tree, even though the normal role of myelin (to speed conduction of the nerve impulse) seems irrelevant here because none of the axonal branches are very long.Previous work by the Angulo group showed that disrupting the normal development of this myelin led to a weakening of the inhibition that these cells generate in excitatory pyramidal cells and thus led to cognitive changes. The current paper disrupts myelin in older animals by a different method – cuprizone – and also finds that this results in reduced inhibition, in part by increasing the cell capacitance and thus making the cell harder to excite, and in part by decreasing the number of release sites for the inhibitory transmitter GABA. Importantly the paper then goes on to assess how this affects the theta and γ rhythms generated in the cortex, and to show that the reduction of inhibition promotes epileptic seizures. The mechanisms are assessed using histology and immunohistochemistry (to quantify myelination, axon structure and transmitter release sites), electrophysiology (to assess synapse strength and cell spiking behavior), and optogenetics (to control the spiking of the cells).The conclusions of this paper are mostly well supported by data, but the following points need to be clarified.(1) For the reader to fully understand the effects on theta and γ rhythms, it would be useful to briefly summarise somewhere the circuitry that produces these oscillations.

We agree this is important information and we have updated the text in the revised version. Regarding γ rhythms we write line 241: “Experimental and computational studies show that in most cortical areas γ rhythms are strongly shaped by electrically and synaptically coupled PV^+^ interneurons which by temporally synchronizing firing rates, the synaptic inhibitory time constants (≅ 9 ms) and the recurrent excitatory feedback from PNs, giving rise to network resonance in the 30 to 80 Hz bandwidth”. Regarding theta rhythms “Physiological theta oscillations are strongly driven by the long-range corticothalamic circuitry and cortical PN firing during non-rapid eye movement (NREM), also called thalamocortical spindles” (line 294)

(2) For clarity: state in the methods, and also early in the main text, that data are (I assume, from reading Table S1 legend) displayed as mean+/-s.e.m. Give exact p values everywhere instead of stars, and state whether correction for multiple comparisons has been made where appropriate.

We agree with the reviewer that clarity about the data and statistical analyses is essential. We included in our original submission a complete overview of all statistical tests and the exact P values for all our data analyses in Table S2. We always used multiple comparisons where required (i.e. when more than two groups were included in one analysis). In the revision, we have now provided a single Excel file that besides the statistical test details with exact values also shows the raw data (Source Data 1) and have added statistical test details in the Figure Legends or main text, providing the exact test values for the stars in the Figures.

(3) State the age of the animals at the start of the results (not just in the methods) because the effects of demyelination may be age dependent (cf the Benamer and Angulo (2020) paper).

For all our demyelination experiments we used mice that were 7–9 weeks old at the start of cuprizone treatment. We have now added this information also at the start of the Results (Line 76). In the article of Benamer et al., 2020 there is no demyelination experiment performed so we are unsure how to address this point.

(4) In Figure 1e-f, contrary to the text it appears that with cuprizone γ power is increased more than 2-fold in quite wakefulness, and that theta power increases 50% when moving. Please give exact p values for these comparisons (and for the theta comparison in quiet wake), and state whether they are corrected for multiple comparisons.

As mentioned above, we should have made it clearer that a complete overview of statistical comparisons was available in Table S2 (now Source Data 1). We do not find statistical support for the notion that theta power is higher during locomotion (dotted lines) or that γ is increased during cuprizone treatment in the wake state (continuous lines). The two-way ANOVA analysis revealed that during quiet wakefulness the power is overall higher in the cuprizone treatment (ANOVA treatment *P* < 0.0001) but not specific to a bandwidth (interaction *P* = 0.0821). Accordingly, post-hoc comparisons also revealed no differences between conditions. In the revied manuscript, the exact P values are provided in the figure legends. We used Šidák’s or Tukey’s post hoc tests to correct for multiple comparisons (see new Source Data 1).

(5) Showing that Shiverer mice also have seizures and interictal discharges is not really a robust test of the concern that (line 102) “interictal epileptiform discharges are due to off-target effects of cuprizone treatment”. The authors could provide information on what happened to the γ and theta power in the MBP- mice, and also state whether cuprizone affected EPSCs in pyramidal cells or interneurons.

Thank you for raising these valuable points. Analyzing the power of distinct bandwidths in the Shiverer mice was unfortunately problematic since we did not have an EMG electrode installed to identify the behavioral state of the animals. As an alternative, we have now analyzed the frequency band power in periods just before and after interictal spikes, to enable a direct comparison with rhythms in cuprizone-treated mice. The new analysis shows that the power across the four frequency bands around interictal spikes was similar between the two groups (Two-way ANOVA Frequency bands x model *P* = 0.7875, *n* = 6 mice, Figure 1—figure supplement 2).

In the original manuscript, we showed that miniature EPSCs from PV interneurons were not different in amplitude or frequency (in the revised version Figure 3—figure supplement 4). The lack of difference in excitatory synaptic currents is in line with our previous EPSP recordings from hippocampal and neocortical PNs in the cuprizone model (Ramaglia et al., 2021; Hamada et al., 2015).

(6) In Line 141 it is stated that normally there was 80% myelination along 83 mm of axons analysed. This contrasts dramatically with the statement of Benamer et al., (2020, Nat Comm, see penultimate para in column 2 of page 11) that only 2.5-4% of each axon is myelinated (cf your Figure 2E). Can you explain this apparent difference (e.g. do you mean to state that 80% cells had at least some myelin on their axon)?

We thank the reviewer for raising this issue. We quantified the proportion of axonal myelination by examining large areas within layer 5 and visual selection of PV^+^ axons, possibly biasing our analysis to larger diameter axons which are typically myelinated (>0.4 µm, Stedehouder et al., 2019). These results also may include long-range PV axons within the neocortex. To the best of our knowledge, Benamer et al., based the 2.5–4% on their experimentally measured total myelinated length combined with a literature value (10 to 22 mm; Koelbl et al., 2019). Using our detailed and extensive morphological analysis of single biocytin-filled PV cells stained for MBP we determined the percentage of PV axon myelination to be on average 2.8 ± 1.2% (n = 4 axons). We have added this estimate to the manuscript (line 133).

(7) Is it possible to define whether the main effect on function is due to a presumed lower firing rate of FSIs in vivo (due to their increased capacitance) or due to their 3-fold weaker IPSCs?

The reviewer raises an excellent question. In comparison to the 2.0-fold reduction in the optical IPSC peak amplitude, the threshold for action potential firing (rheobase) was increased by 1.4-fold. Although both aspects will effectively limit inhibition, our optogenetic rescue experiment showed that when we extrinsically drove BC action potential firing, γ entrainment was entirely abolished (Figure 5). Although this shows the impact of a lack of fast inhibition, it does not inform us whether it is more important in comparison to the increased threshold to action potential generation under in vivo conditions.

(8) Does the change in axon topology reported by Benamer et al., (2020 Nat Comm) contribute to the decreased probability of FSI-SSC connections being detected (see line 218)?

The reviewer asks whether we can explain the results published by Benamer et al., regarding their finding of an absence of connectivity between fast spiking interneurons (FSI) and spiny stellate cells (SSCs). However, we did not collect data that might offer insight into their findings. Moreover, the study of Benamer et al., has a different experimental design than ours since their FS interneuron axons remained myelinated. In our experience, with a complete loss of myelin in adult mice, we did not find differences in the topology of identified PV axons determined over millimeters of reconstructed axons (Figure 2).

(9) In Lines 650-657 it is not really clear how, in the Syt2 analysis of Figure 3 and Figure S4, it was determined whether each of the many boutons visible in control conditions was associated with the axon or soma being studied. Picking them "by hand" seems a bit vague. When you state "colocalization of the pyramidal cell and PV/Syt2" the results obtained will depend critically on the thresholding applied for each label. More details on this are needed so that the reader can implement an objective and identical procedure if needed.

We agree this was insufficiently detailed. We have changed the method description (from line 609 and onwards):

“The intensity of PV^+^ or Syt2 immunostaining was measured with a z-axis profile, calculating the mean RGB value for each z-plane. When quantifying the axosomatic puncta, the soma was defined to extend into the apical dendrite maximally ~4 μm and a boundary was drawn around the maximum edges of the region of interest (ROI). For counting apical dendritic puncta, a 200 µm length of apical dendrite was selected as ROI. Linear immunofluorescent signals from ßIV-spectrin were identified as AIS and used as ROI. For all analyses, the RGB images were separated into single colour channels using the colour deconvolution plugin in Image J. The single-colour channel containing boutons signals was subjected to thresholding and particle filter of 0.5 μm. The threshold was saved and applied to all images in the same staining group. The boutons were selected by scanning through the 3D projection of ROI with 0.35 µm z-steps. Trained experimenters identified the boutons either by colocalization of the ROI and PV/Syt2 or direct contact of the two. The boutons were characterized as round spots with a minimal radius of 0.5 μm ranging to almost 2 μm. Three experimenters blinded to the identity of the experiment group independently replicated the results. All image analysis was done in Fiji (ImageJ) graphic software (v.2.0.0-rc-65/1.5w, National Institutes of Health)”.

(10) Line 261 and following. State the frequency of optical excitation used. How does the γ entrainment depend on the frequency used? Is 30Hz stimulation needed, or would tonic inhibition work as well?

For acute slice recordings, we used 1-second-long pulses of blue light for field illumination. We have not tested different optical stimulation frequencies in vivo. However, Cardin et al., 2009, showed that the peak of entrainment of the LFP to γ occurs at 50 Hz, when testing 10 Hz steps between 10 and 100 Hz. We are not aware of experiments that have examined whether tonic inhibition would work as well, but this could of course be technically challenging given the risk of depolarization block with prolonged continuous optogenetic stimulation.

(11) Line 365. "Interestingly, in contrast to dampening theta oscillations driving myelin-deficient PV+ interneurons at γ-frequencies did not entrain local field oscillations." Can the authors explain why not? Is this because, even if action potentials are generated, the decrease in Syt2+ release sites results in insufficent GABA release onto the pyramidal cells?

Yes, we agree with this viewpoint. Because of losing presynaptic PV terminals, or lacking synchronization of release sites, the reduced fast component in PV-mediated inhibition should limit the capacity for γ entrainment. See also our answer to point 7 above.

(12) In Figure 5A legend, does the statement "Blue light was switched on during high interictal activity (>10 spikes/min)." apply to all of the rest of the figure? If so, say so.

Thank you for pointing this out. In figure 5A, the blue light was switched “on” only during high interictal activity (>10 spikes/min). For optical γ entrainment (Figure 4), we used 1-ms duration pulses of blue light, repeated forty times, switched randomly during states of awake locomotor activity.

(13) Line 379: Shouldn't "hypermyelinated" read HYPOmyelinated"?

We have changed the wording to “aberrant myelination” since there was neither evidence for less nor for more myelination of fast-spiking interneurons (see their Supplementary Figure 2 of Benamer et al., 2020).